# Dendritic cells maintain anti-tumor immunity by positioning CD8 skin-resident memory T cells

Jennifer L Vella[1], Aleksey Molodtsov[1], Christina V Angeles[2], Bruce R Branchini[3], Mary Jo Turk[1,4], Yina H Huang[1,4,5]

**Tissue-resident memory ($T_{RM}$) T cells are emerging as critical components of the immune response to cancer; yet, requirements for their ongoing function and maintenance remain unclear. APCs promote $T_{RM}$ cell differentiation and re-activation but have not been implicated in sustaining $T_{RM}$ cell responses. Here, we identified a novel role for dendritic cells in supporting $T_{RM}$ to melanoma. We showed that CD8 $T_{RM}$ cells remain in close proximity to dendritic cells in the skin. Depletion of CD11c$^+$ cells results in rapid disaggregation and eventual loss of melanoma-specific $T_{RM}$ cells. In addition, we determined that $T_{RM}$ migration and/or persistence requires chemotaxis and adhesion mediated by the CXCR/CXCL16 axis. The interaction between CXCR6-expressing $T_{RM}$ cells and CXCL16-expressing APCs was found to be critical for sustaining $T_{RM}$ cell–mediated tumor protection. These findings substantially expand our knowledge of APC functions in $T_{RM}$ T-cell homeostasis and longevity.**

## Introduction

Tissue resident memory T ($T_{RM}$) cells are a unique subset of memory cells persisting at initial sites of challenge ([1], [2]) and are important in memory responses against viruses, bacteria, fungi and parasites ([3]). Unique properties including permanent residence in tissue and constitutive expression of inflammatory cytokines and effectors make $T_{RM}$ cells ideal sentinels acting as first responders to reinfection of pathogens ([4]). Since their discovery, $T_{RM}$ cells have been identified within multiple epithelial barriers and within sites of immune privilege including the brain ([5]). Whereas $T_{RM}$ cells share a core transcriptional signature, tissue-specific transcripts induced by differential activity of Blimp-1, Hobit, and Runx3 confer unique properties required for their site-specific persistence ([6]). Given the importance of $T_{RM}$ cells in protective immunity, it is critical to identify mechanisms supporting their persistence. $T_{RM}$ cells adapt to their location by using local fuel sources ([7]) indicating that they rely heavily on the tissue environment for their long-term maintenance, leading us to determine whether specific accessory cells maintain $T_{RM}$ cell homeostasis within a tissue niche.

The immune response to solid cancers is limited by availability of neoantigens, tolerance to self-antigens, T-cell dysfunction, and immunosuppression within the tumor microenvironment. Checkpoint blockade immunotherapy, which blocks inhibitory receptors such as PD-1 and CTLA-4, can reverse T-cell dysfunction and when combined achieves >50% patient responsiveness. However, predicting patient responders remains a challenge. Recently, expression of $T_{RM}$ cell markers in tumor infiltrating lymphocytes has been reported to correlate with patient responsiveness and prolonged survival in various solid cancers including breast, lung, pancreatic, and melanoma ([8], [9], [10], [11]), suggesting that $T_{RM}$ cells mount effective anti-tumor responses. It is also well recognized that melanoma patients who develop spontaneous or immunotherapy-induced autoimmune vitiligo achieve longer progression-free survival ([8], [9]). Using a melanoma-associated vitiligo (MAV) mouse model, we recently demonstrated that after tumor removal, melanocyte-specific CD8 $T_{RM}$ cells develop within the skin of vitiligo-affected mice and are necessary and sufficient for durable anti-melanoma protection ([12]). Together, these data support an emerging theme that development and continued maintenance of tumor reactive $T_{RM}$ cells is critically important for immunity against tissue tumors. Using the MAV model, we sought to identify factors that trigger and/or sustain tumor reactive $T_{RM}$ cells. Identification and characterization of $T_{RM}$ cell requirements may reveal new strategies to promote curative effects in cancer patients.

$T_{RM}$ cells were originally identified in the context of HSV skin infection co-expressing CD69 and CD103 ([13]). It is now apparent that they are a heterogenous population and cannot be defined by two proteins alone ([14], [15]). Through RNA-sequencing analyses of virally induced skin $T_{RM}$ cells, a core signature of genes has been identified and includes the chemokine receptor CXCR6 (Bonzo) ([16]). CXCR6 interacts exclusively with CXCL16, a chemokine that is secreted or expressed on the surface of a number of cell types including DCs, myeloid cells, epithelial cells, and keratinocytes ([17], [18], [19], [20]). Although early studies implicate CXCR6 in recruitment of natural killer T cells and T cells to various autoimmune sites ([21], [22]), CXCR6

[1]Department of Microbiology and Immunology, Geisel School of Medicine at Dartmouth, Lebanon, NH, USA   [2]Department of Surgery, University of Michigan, Rogel Cancer Center, Ann Arbor, MI, USA   [3]Department of Chemistry, Connecticut College, New London, CT, USA   [4]Norris Cotton Cancer Center, Lebanon, NH, USA   [5]Department of Pathology and Laboratory Medicine, Dartmouth Hitchcock Medical Center, Lebanon, NH, USA

Correspondence: yina.h.huang@dartmouth.edu

is also important for the recruitment of CD8 T cells in response to viral and bacterial infections ([23], [24]). CXCR6 recruits and positions CD8 $T_{RM}$ cells proximal to CXCL16-expressing airway epithelia after influenza infection ([25]). However, lung epithelia associated $T_{RM}$ cells are short-lived and require constant CXCR6-dependent replenishment by self-renewing $T_{RM}$ cells resident within the lung interstitium ([26]). Recently, CXCR6 was also reported to support migration of CD8 $T_{RM}$ cells to the lung after tumor vaccination ([27]). Thus, the CXCR6/CXCL16 axis plays a traditional role in the chemotaxis of both effector cells and terminally differentiated $T_{RM}$ cells. Unique biochemical properties of CXCR6 and transmembrane-bound CXCL16 also confer an unconventional role for the pair in mediating cell–cell adhesion between activated T cells and DCs ([28]). CXCR6 is the only chemokine receptor that encodes a DRF amino acid sequence motif instead of the consensus DRY motif in its cytoplasmic domain that diminishes migration while increasing adhesion ([29]). However, it remains unclear whether CXCR6 is required for the establishment and/or maintenance of $T_{RM}$ cells in the skin, what role it has in tumor immunity, and whether its role includes chemotaxis and/or adhesion.

In this study, we report an important requirement for CXCR6 expression on CD8 $T_{RM}$ cells in maintaining their long-term skin residence through direct aggregation with transmembrane CXCL16-expressing DCs. Short-term depletion of CD11c⁺ cells led to a reduction in skin CXCL16 expression and CD8 $T_{RM}$ cell dispersal, whereas longer depletion led to $T_{RM}$ cell loss. While CXCR6-deficient CD8 effector T cells were capable of infiltrating primary tumors, CD8 $T_{RM}$ cell positioning within MAV-affected skin was disrupted, resulting in defective anti-tumor memory responses. Together, these data define a new role for antigen presenting cells in capturing and holding CD8 $T_{RM}$ cells within peripheral tissues through CXCL16/CXCR6-mediated migration and/or adhesion.

## Results

### CD8 $T_{RM}$ cells form large clusters with CD11c⁺ cells

We previously established a model of MAV based on depletion of regulatory T cells that results in hair depigmentation in 60–70% of wild-type mice (Fig 1A) ([30], [31]). CD8 $T_{RM}$ cells are present in abundant numbers in the skin of depigmented (MAV affected) but not unaffected mice ([12]). Moreover, >95% of the CD8 T cells isolated from the skin of MAV-affected mice display phenotypic markers indicative of $T_{RM}$ cells (Fig S1A). Using immunofluorescence microscopy, we confirmed that CD8 $T_{RM}$ cells localized at the dermal-epidermal junction and near hair follicles of MAV-affected skin but were infrequent in unaffected skin ([12]). In addition, we observed large clusters of CD103⁺KLRG1⁻ CD8 $T_{RM}$ cells surrounding the base of hair follicles in MAV affected but not unaffected skin (Figs 1B–E and S1B and C). The appearance of skin CD8 T-cell clusters was reminiscent of tertiary lymphoid structures (TLSs) reported in the context of certain cancers, autoimmune diseases, and infections ([32], [33], [34], [35], [36]). To determine whether $T_{RM}$ cell clusters were TLS, MAV-affected skin was examined for the presence of cell types and organizational structures characteristic of TLS. Skin and skin-draining LNs from MAV-affected mice were stained with antibodies to detect B cells, high endothelial

venules and T cells. We found that in contrast to LNs, CD8 $T_{RM}$ cell aggregates contained little to no B cells, high endothelial venules, or CD4 T cells (Fig S2A and B) indicating that hair follicle–associated $T_{RM}$ cell clusters in MAV-affected skin are distinct structures from TLS.

While the observed clusters are not TLS, T cells are known to cluster with APCs in response to various infections ([37], [38], [39], [40]). To determine whether skin CD8 $T_{RM}$ cells co-clustered with APCs, MAV-affected and unaffected skin sections were stained with antibodies specific for CD11c, a marker expressed on a number of APCs including DCs and macrophages ([41]). Immunofluorescence microscopy showed an abundance of CD11c⁺ cells clustering in near equal numbers with CD8 $T_{RM}$ cells in MAV skin (Fig 2A and B). The distance between CD8 T cells and the nearest CD11c⁺ cell was significantly shorter for skin from MAV-affected mice than control unaffected mice supporting the notion that CD8 T cells interact with CD11c⁺ cells (Fig 2C). We next sought to determine whether similar clusters form in the skin of patients with MAV. Immunohistochemical staining of skin sections from normal donors and sites of depigmentation distal to primary tumors from MAV patients identified higher numbers of CD8 T-cell clusters in MAV patients (Fig 2D and E). Interestingly, human CD8 T cells and CD11c⁺ cells in depigmented skin resided in both follicular and interfollicular regions that bordered pigmented skin, identified by Fontana Masson staining (Fig S3). The total cell numbers of CD11c⁺ cells were not significantly different; however, CD11c⁺ cells were in close proximity to CD8 T cells in skin from MAV patients compared with control skin (Fig 2E and F). Altogether, these data indicate that skin $T_{RM}$ cells cluster with CD11c⁺ cells in skin of mice and human MAV patients.

### CD11c⁺ cells express CXCL16 in MAV-affected skin

Multiple CD11c⁺ macrophages and DC subsets reside in lymphoid and non-lymphoid compartments. To broadly distinguish macrophages from DCs in MAV skin, we examined F4/80, CD11b, and CD11c expression by immunofluorescence microscopy. The macrophage marker F4/80 was not detected on cells within the clusters; however, differential CD11b expression distinguished two CD11c⁺ populations: CD11c⁺CD11b^neg and CD11c⁺CD11b⁺ (Figs 2G and H and S4A and B).

The close proximity of CD8 $T_{RM}$ cells and CD11c⁺ cells prompted us to determine whether specific receptor–ligand pairs or chemokine–chemokine receptors were involved in coordinating their interaction. To identify putative interacting proteins, we examined the transcriptional profiles of CD8 $T_{RM}$ cells and both CD11c⁺ subsets using RNA-sequencing analysis. Bulk CD8 T cells and Ag-specific Pmel CD8 T cells were isolated from MAV-affected skin and found to express $T_{RM}$ cell–specific genes distinct from naïve CD8 T cells (Fig S1C). CD11c⁺⁻CD11b^neg and CD11c⁺CD11b⁺ cells expressed distinct myeloid-enriched genes (Fig S4C and D). CD11c⁺CD11b^neg cells expressed a gene signature more closely associated with conventional type 1 DCs, whereas CD11c⁺CD11b⁺ cells expressed genes found in multiple myeloid populations, which may be indicative of a mixed population (Fig S4E).

Because chemokines play important roles in the organization and maintenance of higher order lymphoid structures ([35], [36]), we were particularly interested in the expression of chemokines and their complementary receptors by CD11c⁺ cells and CD8 $T_{RM}$ cells. We found that chemokines that are induced during inflammation,

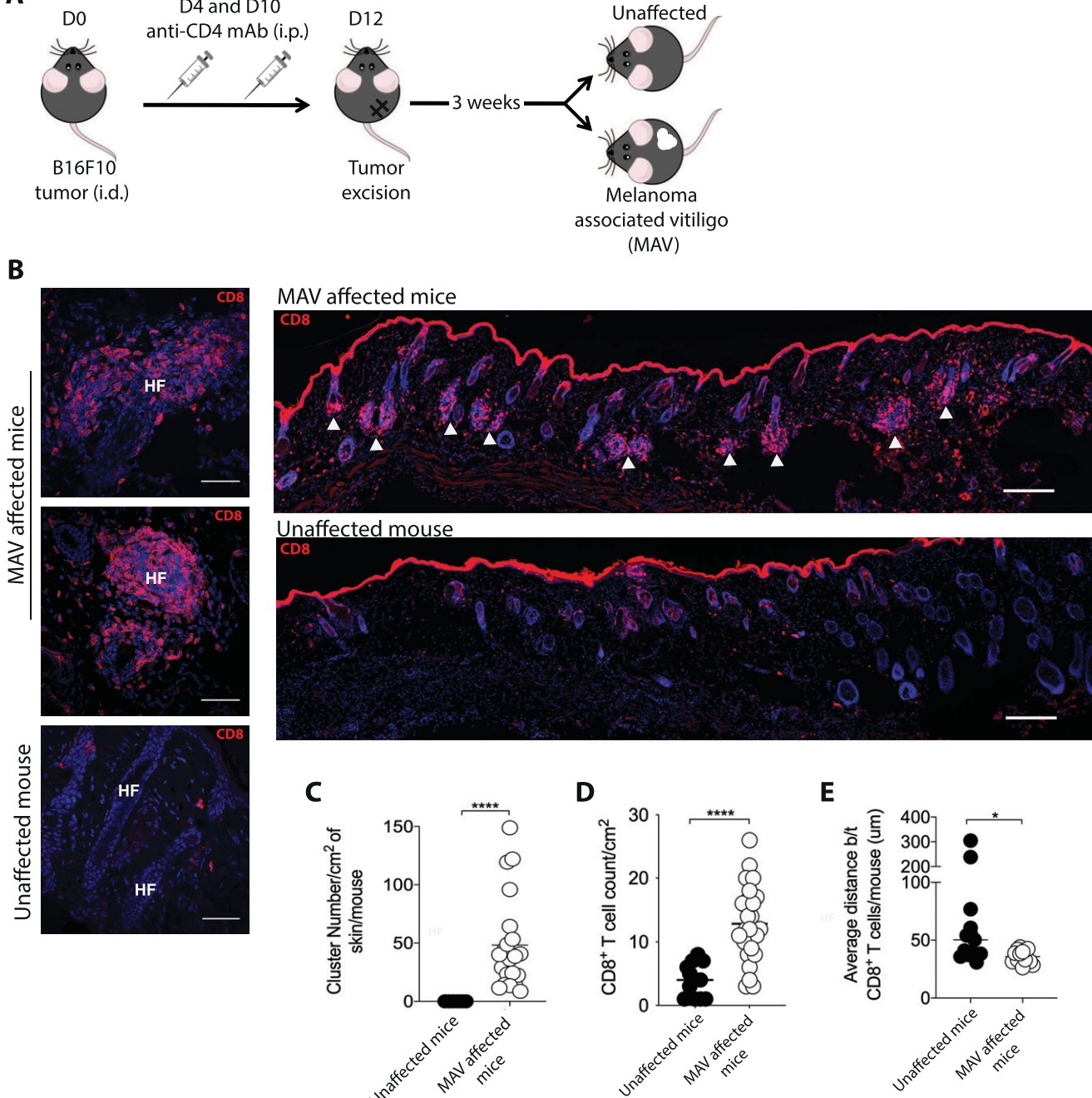

**Figure 1. CD8 T$_{RM}$ cells cluster around hair follicles in melanoma-associated vitiligo (MAV)–affected skin.**
**(A)** Experimental outline to induce MAV. Unaffected mice underwent the same procedure but did not present with depigmentation; skin was harvested from the surgical site of mice >30 d post-surgery unless otherwise indicated. **(B)** CD8$\beta^+$ T cells clustered around hair follicles in skin of MAV-affected mice but not in skin of unaffected mice. Stains identify CD8$\beta$ (red) and nuclei (blue). HF, hair follicle, white arrows indicate clusters. Scale bar, 50 and 100 $\mu$m. **(C)** Number of CD8 T-cell clusters found in skin from MAV-affected and unaffected mice. **(D)** Number of CD8 T cells found in skin from MAV-affected and unaffected mice. **(E)** Average distance between CD8 T cells in skin from the MAV-affected and unaffected mice. **(C, D, E)** n = 6–25 mice pooled from five independent experiments (C, D); n = 6–8 mice pooled from three independent experiments (E). **(C, D, E)** Symbols represent individual clusters (C), CD8 T-cell count per mouse (D), or average distance between CD8 T cells per mouse (E), horizontal lines indicate mean. Significance was determined by Mann–Whitney test; *$P$ = 0.0145, ****$P$ ≤ 0.0001.

CXCL9 and CXCL16, were expressed by both CD11c$^+$ subsets, whereas their respective receptors CXCR3 and CXCR6 were expressed by CD8 T$_{RM}$ cells (Fig 3A). It is well established that CXCL9 and CXCL10 are induced by IFNγ and function to promote effector T cell infiltration into tumors as well as sites of active depigmentation in the setting of vitiligo (42, 43). However, we found that the establishment of MAV was unaffected by CXCR3 deficiency (Fig S5) leading us to consider the importance of CXCL16 and CXCR6 in CD8 T$_{RM}$ cell organization in

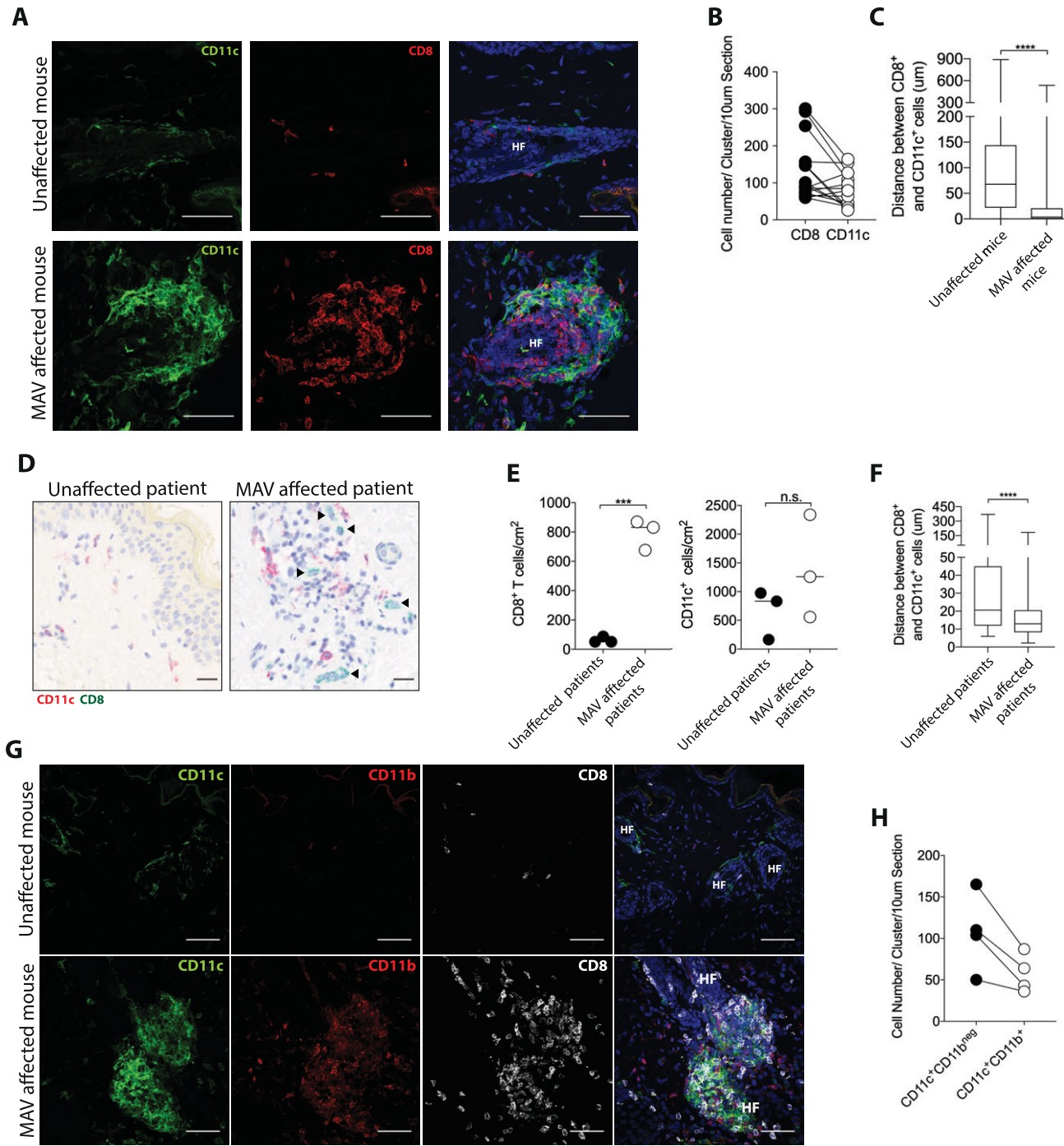

**Figure 2.   CD8 tissue-resident memory cell clusters contain CD11c+ cells.**

**(A)** CD8 T_RM cells and CD11c+ cells cluster together in skin from melanoma-associated vitiligo (MAV)–affected mice but not in unaffected skin, CD8β (red), CD11c (green), and nuclei (blue). Scale bar, 50 μm. **(B)** Number of CD8 T cells and CD11c+ cells per cluster in a 10 μm thick skin section; n = 15 clusters pooled from three independent experiments. **(C)** Distance between CD8 T cells and CD11c+ cells in skin from unaffected and MAV-affected mice; n = 6–8 mice pooled from three independent experiments. **(D)** CD8 T cells and CD11c+ cells in skin from unaffected and MAV patients. CD8 (green) and CD11c (red). Scale bar, 25 μm. Black arrows indicate CD8 T cells. **(E)** Number of CD8 T cells and CD11c+ cells found in skin from unaffected and MAV-affected patients. **(F)** Distance between CD8 T cells and CD11c+ cells in skin from unaffected and MAV-affected patients. **(G)** CD8 T cells, CD11c+ cells, and CD11b+ cells in skin from unaffected and MAV-affected mice. CD8β (white), CD11c (green), CD11b (red), and nuclei (blue). Scale bar, 50 μm. **(H)** Quantification of CD11c+CD11b^neg and CD11c+CD11b+ cells per cluster in a 10 μm thick skin section; n = 4 clusters representative of three independent experiments. **(B, E)** Symbols represent individual clusters (B) or individual patients (E). Horizontal lines indicate mean. **(C, E, F)** Significance was determined by Mann–Whitney test; ****$P \leq 0.0001$ (C, F), unpaired $t$ test; ***$P = 0.0003$ (E).

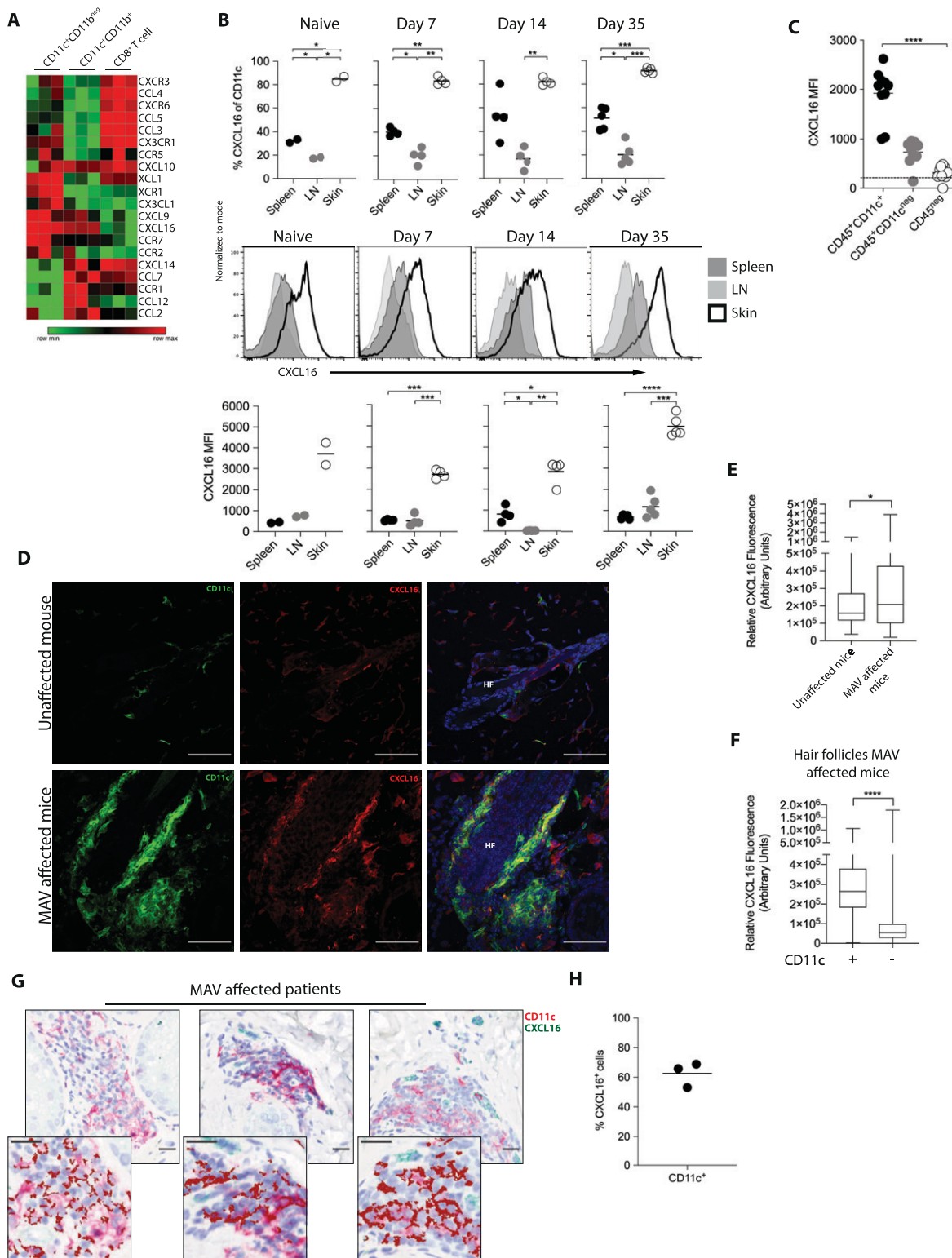

**Figure 3. CD11c⁺ cells express CXCL16 in melanoma-associated vitiligo (MAV)–affected skin.**

**(A)** Heat map showing gene expression of chemokines and receptors in CD11c⁺CD11b^neg, CD11c⁺CD11b⁺, and CD8⁺ T cells sorted from skin of MAV-affected mice. Data presented as log2-normalized expression. **(B)** Surface expression of CXCL16 on CD11c⁺ cells isolated from skin of naïve (n = 3) or MAV-affected (n = 5) mice, skin-draining LNs, or spleen 7, 14, and 35 d after surgery. Representative of three independent experiments. **(C)** Surface expression of CXCL16 on CD45⁺CD11c⁺, CD45⁺CD11c^neg, and CD45^neg cells isolated from skin of MAV-affected mice, 35 d after surgery. Representative of two independent experiments with skin collected between 35- and 60-d post-surgery. Dotted horizontal line indicates background CXCL16 expression in unstained cell sample. **(D)** Expression of CXCL16 and CD11c in unaffected and MAV-affected mice; CD11c

the skin. Unlike traditional chemokines, CXCL16 is a multifunctional protein that acts not only as a chemoattractant but also as a cell surface expressed adhesion molecule and scavenger receptor on APCs ([18], [44]).

The juxtaposition of CD8 T cells and CD11c[+] cells surrounding the hair follicles led us to examine surface expression of CXCL16 on CD11c[+] cells isolated from skin, skin-draining LN, and spleen of MAV-affected and naïve mice. Approximately 90% of all CD11c[+] cells isolated from skin of naïve or MAV-affected mice expressed surface CXCL16 compared with 10–60% of CD11c[+] cells from spleen and skin-draining LN ([Fig 3B]). CXCL16 mean fluorescence intensity (MFI) was increased on CD11c[+] cells isolated from both naïve or MAV skin compared with those isolated from spleen and skin-draining LN ([Fig 3B]). In addition to CD11c[+] cells, CXCL16 is constitutively expressed by keratinocytes in the healthy epidermis and secreted upon exposure to inflammatory cytokines including TNF-$\alpha$, IFN-$\gamma$, and IL-1$\beta$ ([19], [45]). To identify the cell types expressing CXCL16 in the skin, we performed flow cytometry and immunofluorescence on the skin from MAV-affected and unaffected mice. Results generated through flow cytometry revealed that CXCL16 expression was low on CD45[neg] cells compared to CD45[+] leukocytes. However, CD11c[+] cells expressed the highest levels of surface CXCL16 on a per cell basis compared to CD11c[neg] leukocytes ([Fig 3C]). As previously shown, lymphoid clusters are densely associated with hair follicles of MAV mice. CXCL16 immunofluorescence identified total surface and secreted CXCL16 expression was enriched at hair follicles containing CD8 T cell and CD11c[+] cell clusters. An overall comparison of CXCL16 fluorescence within hair follicles revealed that skin from MAV-affected mice expressed significantly more CXCL16 in comparison to unaffected mice ([Fig 3D and E]). In addition, CXCL16 expression level was significantly higher surrounding hair follicles containing CD11c[+] cells compared with hair follicles lacking CD11c[+] cells ([Fig 3F]), indicating that the presence of CD11c[+] cells is associated with CXCL16 expression. Immunohistochemical analyses of skin from multiple human MAV patients also showed CXCL16 expression around leukocyte aggregates ([Fig 3G]). Moreover, 60% of CXCL16 expression co-localized with CD11c[+] cells ([Fig 3H]). Together, these data indicate that CD11c[+] cells are the predominant source of CXCL16 within CD8 T[RM] cell clusters in MAV skin.

## Depletion of CD11c[+] cells results in a reduction in CD8 T[RM] cells in skin

To determine whether CD11c[+] cells are required to maintain CD8 T[RM] cell clusters in MAV skin, we used mice expressing the diphtheria toxin (DT) receptor controlled by the CD11c promoter (CD11c.DTR) for which administration of DT selectively depletes CD11c[+] cells. MAV was established in CD11c.DTR[+] and CD11c.DTR[neg] littermate mice followed by three doses of DT over the course of 7 d to determine the effects of short-term CD11c depletion on CD8 T[RM] cell clusters

([Fig 4A]). A complete loss of CD11c[+] cells was observed in skin from DT treated CD11c.DTR[+] MAV-affected mice but not from CD11c.DTR[neg] littermates ([Fig 4B]). Short-term CD11c depletion also resulted in a significant reduction in the number of CD8 T[RM] cell clusters ([Fig 4C]). In addition, there was a reduction in CXCL16 fluorescence intensity after CD11c depletion indicating that loss of CD11c[+] cells abrogated CXCL16 expression even by CD11c[neg] cells ([Fig 4D and E]).

Whereas DT treatment in CD11c.DTR[+] mice resulted in a reduction in clusters, the percentage of CD8 T cells in MAV skin did not differ between CD11c.DTR[neg] and CD11c.DTR[+] mice ([Figs 4C] and [S6A and B]). Therefore, we next assessed the effects long-term depletion of CD11c[+] cells have on the CD8 T[RM] cell population in MAV skin. Because long-term DT treatment of CD11c.DTR mice results in neurotoxicity, bone marrow chimeric mice were generated by reconstituting lethally irradiated wild-type congenic CD45.1 mice with CD11c.DTR bone marrow. 6 wk post reconstitution, MAV was established in bone marrow chimeric mice followed by DT or PBS treatment every 3 d for 30 d ([Fig 4F]). Long-term DT treatment resulted in a significant reduction in the proportion of CD8 T cells and CD11c[+] cells in MAV skin ([Fig 4G]). Altogether, these data have identified a requirement for CD11c[+] cells in producing CXCL16 and organizing and maintaining CD8 T[RM] cell residence within the skin.

## CXCR6 is up-regulated and sustained in MAV skin

To determine the importance of CXCL16 interaction with CXCR6 on CD8 T[RM] cells in the skin, we first examined CXCR6 expression by CD8 T cells at various stages during MAV development. Because MAV develops in only 60–70% of wild-type mice and becomes apparent 30 d after tumor excision, we tracked antigen-specific T cell responses by adoptively transferring Thy1.1-marked CD8 T cells expressing the Pmel T cell receptor that recognizes gp100[25–33] melanocyte antigen and pre-activated to allow retroviral expression of PpyRE9 luciferase reporter gene (Luc[+]Pmel) ([Fig 5A]). Luc[+-]Pmel cells localized to the tumor and tumor draining LN before tumor resection ([Fig 5B]). Beginning 3 d post-surgery an observable increase in luminescence signal was detected in mice that developed MAV, whereas signal continually decreased in unaffected mice ([Fig 5B]). Luc[+]Pmel cells were isolated 7, 14, or 35 d after tumor excision and analyzed by flow cytometry for CXCR6 expression. Because chemokine-binding results in rapid internalization of their receptors, we examine both surface and intracellular expression of CXCR6 ([46]). Total CXCR6 expression was compared between CD8 Pmel T cells isolated from the spleen, draining LN, and skin proximal to the surgical site of mice with high luciferase signal. CXCR6 expression was constitutively expressed in 60–80% of CD8 Pmel T cells isolated from all three sites in naïve mice. However, the proportion of CD8 Pmel T cells expressing CXCR6 increased to 80–100% in each organ after surgery ([Fig 5C]). The percentage of cells expressing CXCR6 reached its peak by day 7 post-

(green), CXCL16 (red), and nuclei (blue). Scale bar, 50 $\mu$m. **(E)** CXCL16 expression in IF images from skin of unaffected and MAV-affected mice; n = 5–8 mice. **(F)** CXCL16 expression in hair follicles with or without CD11c[+] cell clusters in MAV-affected skin; n = 8 mice. **(G)** CD11c[+] and CXCL16[+] cells in skin from MAV-affected patients. Stains identify CXCL16 (green) and CD11c (red); dark red identifies colocalization of CD11c and CXCL16. Scale bar, 25 $\mu$m. **(H)** Percent of CD11c[+] cells expressing CXCL16 in MAV-affected patient skin. **(B, C, H)** Symbols represent individual mice (B, C) or patients (H); horizontal lines indicate mean. MFI, mean fluorescence intensity. Significance was determined by one-way ANOVA; *$P \leq 0.03$, ***<$P = 0.001$, ****$P \leq 0.0001$.

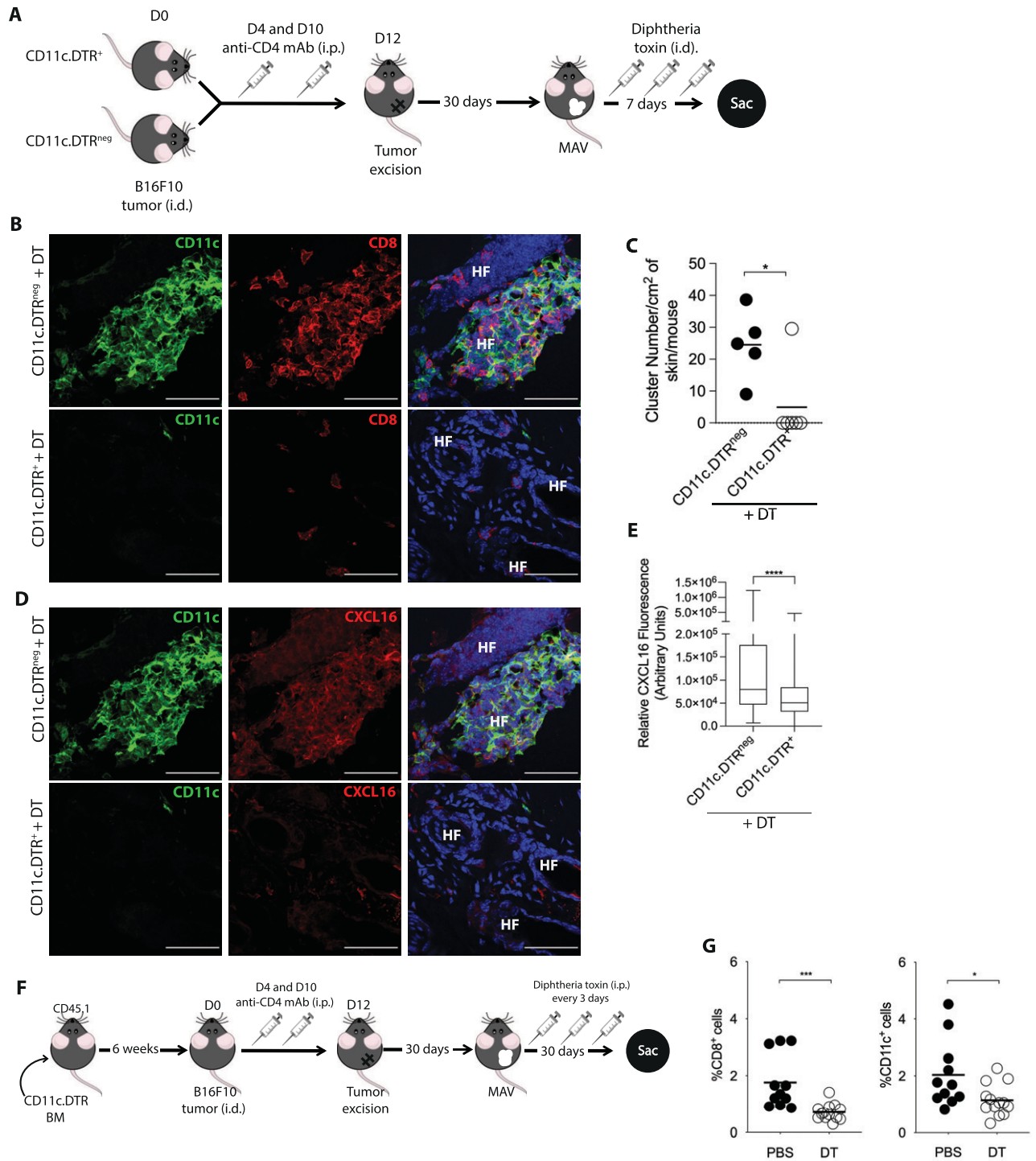

**Figure 4. Depletion of CD11c⁺ cells results in a reduction of CD8 tissue resident memory cells.**
**(A, B, C, D)** Melanoma-associated vitiligo (MAV) was induced in CD11c.DTR⁺ and CD11c.DTRⁿᵉᵍ mice. 30 d after surgery, mice received three doses of 50 ng diphtheria toxin (DT) i.d. over 7 d at the surgical site before the skin was analyzed. **(B)** Expression of CD8 T cells and CD11c⁺ cells in skin from CD11c.DTRⁿᵉᵍ and CD11c.DTR⁺ MAV-affected mice; CD11c (green), CD8β (red), and nuclei (blue). Scale bar, 50 μm. **(C)** Number of clusters found per cm² of skin from CD11c.DTR⁺ and CD11c.DTRⁿᵉᵍ MAV-affected mice treated with DT. **(D)** Expression of CXCL16 and CD11c in skin from CD11c.DTRⁿᵉᵍ and CD11c.DTR⁺ MAV-affected mice treated with DT; CD11c (green), CXCL16 (red), and nuclei (blue). Scale bar, 50 μm. **(E)** Relative CXCL16 expression in skin from CD11c.DTR⁺ and CD11c.DTRⁿᵉᵍ MAV-affected mice treated with DT. **(F, G)** 6 wk after lethally irradiated WT congenic CD45.1 mice received CD11c.DTR bone marrow, MAV was induced and beginning d30 after surgery mice received 250 ng DT or PBS i.p. every 3 d until skin was harvested d60 post-surgery. **(G)** Percent of CD8 T cells or CD11c⁺ cells isolated from skin of MAV-affected mice treated with either DT or PBS. **(C, G)** Symbols represent individual clusters (C), individual mice (G); horizontal lines indicate mean. **(B, C, D, E, G)** n = 4–6 mice pooled from two independent experiments (B, C, D, E) n = 10–14 mice pooled from two independent experiments (G) Significance was determined by Mann–Whitney test; *P ≤ 0.04, ***P ≤ 0.0001.

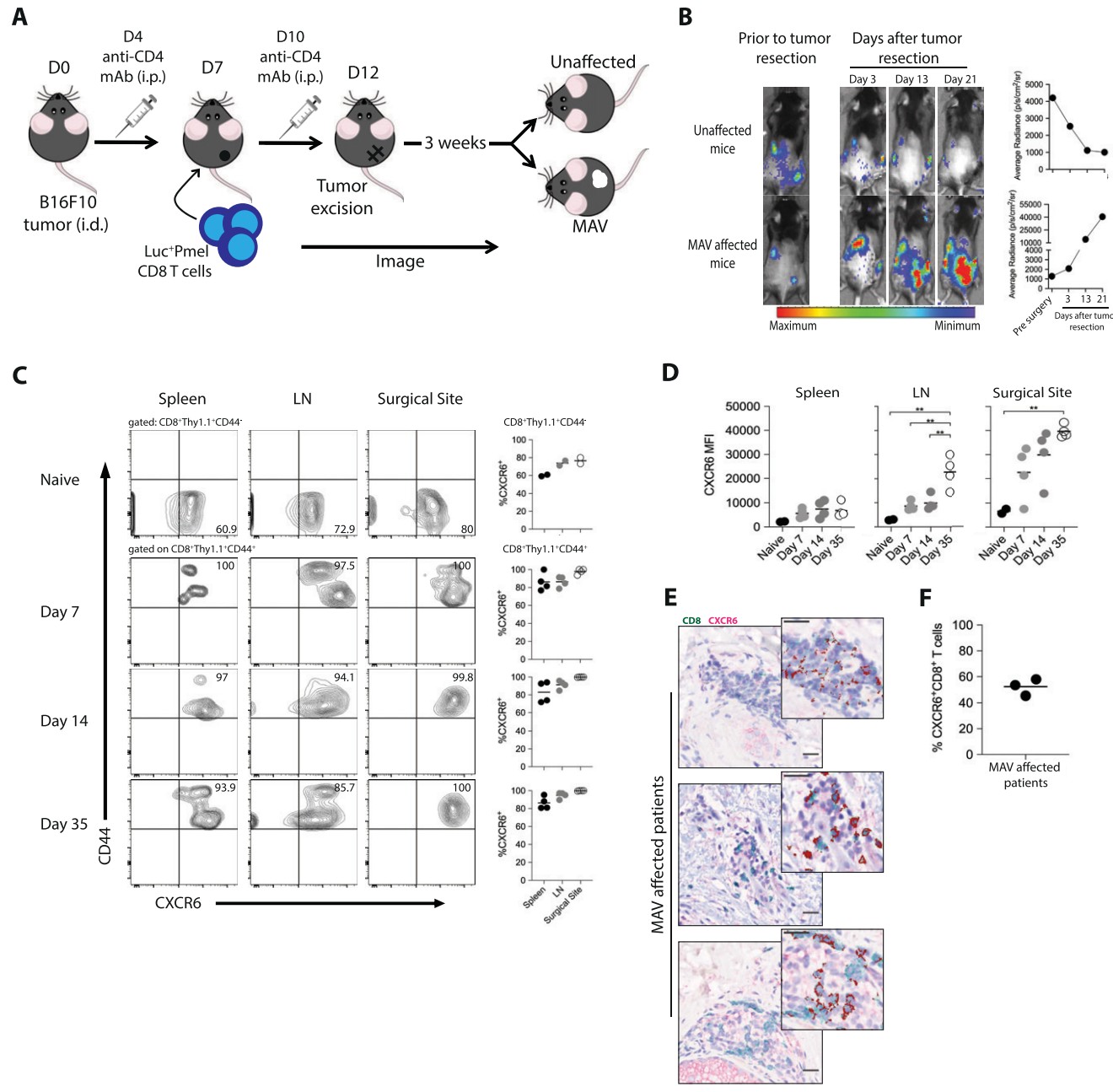

**Figure 5. CXCR6 is up-regulated and sustained in melanoma-associated vitiligo (MAV)–affected skin.**
**(A)** Experimental outline to induce MAV. Mice received 5 × 10⁵ activated Pmel CD8 T cells expressing a luciferase reporter gene (Luc⁺Pmel) 5 d prior to tumor resection.
**(B)** Luminescence signal of Luc⁺Pmel cells over the course of 21 d post tumor excision in unaffected and MAV-affected mice. **(C)** Representative flow plots showing expression of CXCR6 and CD44 on Luc⁺Pmel CD8 T cells. Cells were isolated from experimental mice on days 7, 14, or 35 post-surgery and were identified by CD8⁺Thy1.1⁺CD44⁺; Naïve Pmel CD8 T cells were isolated from naïve mice and identified by CD8⁺Thy1.1⁺CD44⁻. Graphs are compiled data comparing the percentage of CXCR6 expression on CD8⁺Thy1.1⁺CD44⁺ (days 7, 14, and 35) or CD8⁺Thy1.1⁺CD44⁻ (naïve) cells isolated from skin, skin-draining LN, or spleen. **(D)** CXCR6 mean fluorescence intensity expression on CD8⁺Thy1.1⁺CD44⁺ (day 7, 14, 35) or CD8⁺Thy1.1⁺CD44⁻ (naïve) cells isolated from skin, skin-draining LN, or spleen. **(E)** Colocalization of CD8 and CXCR6 staining in skin from MAV-affected patients. Stains identify CD8 (green) and CXCR6 (pink); dark red indicates colocalization of CD8 and CXCR6. Scale bar, 25 μm.
**(F)** Percent of CD8 T cells in skin from MAV-affected patients that are positive for CXCR6. **(C, F)** Symbols represent individual mice (C, D), individual patients (F); horizontal lines indicate mean. **(C, D)** n = 2–4; representative of three independent experiments (C, D). Significance was determined by one-way ANOVA; **P ≤ 0.010.

surgery and remained elevated at least until day 35 post-surgery (Fig 5C). However, the CXCR6 mean fluorescence intensity (MFI) continued to increase on CD8 Pmel T cells isolated from LNs and skin from MAV mice with the highest overall CXCR6 expression observed in skin (Fig 5D). CXCR6 expression on splenic

CD8⁺Thy1.1⁺CD44⁺ cells remained low throughout the experiment (Fig 5D). To determine whether CD8 T cells express CXCR6 in the skin of patients with MAV, skin sections from three different MAV patients were co-stained with CD8- and CXCR6-specific antibodies. We found that ~50% of the CD8 T cells in patient MAV skin expressed CXCR6

(Fig 5E and F). Together, these data indicate that CD8 $T_{RM}$ cells express significant amounts of CXCR6 in MAV-affected mice and patients.

### CXCR6 expression on CD8 $T_{RM}$ cells is required for their long-term maintenance in skin

We have established that CXCR6 is expressed by CD8 $T_{RM}$ cells and that its ligand CXCL16 is expressed on the surface of closely associated CD11c$^+$ cells. To determine whether CXCR6 and CXCL16 interaction is required for leukocyte clustering in skin, MAV was established in wild-type and CXCR6-deficient (CXCR6$^{-/-}$) mice. CD8 effector T cells develop in response to regulatory T cell depletion, infiltrate B16F10 tumors and mediate melanocyte destruction in the skin. Effector T cell dependent melanocyte killing becomes evident in wild-type mice 20–30 d post tumor resection with the regrowth of white hair at the surgical site in 60–70% of treated mice (Fig 6A). In comparison to wild-type, effector CD8 T cells from CXCR6$^{-/-}$ mice also infiltrated B16F10 tumors and promoted depigmentation with similar kinetics and magnitude (Figs 6A and S7A and B). Together, this indicates that CXCR6 is dispensable for CD8 T-cell activation, effector cell recruitment and melanocyte destruction.

CD8 $T_{RM}$ cells develop in the skin of MAV-affected wild-type mice and radiate outward from the tumor excision site disseminating depigmentation 2–3 wk following treatment (Fig 6B). However, in contrast to wild-type mice, depigmentation did not disseminate in CXCR6$^{-/-}$ mice (Fig 6B). Immunofluorescence microscopy and flow cytometric analyses revealed fewer CD8 T cells in MAV skin of CXCR6$^{-/-}$ mice compared with wild-type mice 60 d post-surgery (Fig 6C and D), indicating that CXCR6 is important for the maintenance of CD8 $T_{RM}$ cells. Alternatively, CXCR6 may be required for establishing or disseminating areas of vitiligo.

Our previous findings demonstrated that CD11c$^+$ cells were required for the maintenance of CD8 $T_{RM}$ cells likely through the interactions between CXCR6 and CXCL16. To determine whether there is a cell-intrinsic requirement for CXCR6 in CD8 $T_{RM}$ cell skin localization, we co-transferred naïve WT-Pmel and CXCR6$^{-/-}$-Pmels at a 1:1 ratio into WT recipient mice before inducing MAV. Immunofluorescence microscopy revealed both WT-Pmel and CXCR6$^{-/-}$-Pmels localized along the dermal-epidermal junction (Fig 6E); however, quantification demonstrated a significant reduction in CXCR6$^{-/-}$-Pmel T cells compared to WT-Pmel T cells in MAV-affected skin (Fig 6F). In addition, greater distances were observed between CXCR6$^{-/-}$-Pmel T cells and CD11c$^+$ cells than between WT-Pmel T cells and CD11c$^+$ cells (Fig 6G). Together, these data show that CXCR6 is required for coordinating skin localization of CD8 $T_{RM}$ cells.

### CXCR6 expression on CD8 $T_{RM}$ cells is required for durable tumor immunity

It has been well established that MAV-affected mice are better protected against melanoma tumor re-challenge (31, 47). Only within the last several years, has it been demonstrated that skin-resident memory CD8 T cells are responsible for mediating long lasting anti-tumor protection (12). Our data showing an intrinsic requirement for CXCR6 expression on CD8 $T_{RM}$ cells led us to assess its role in mediating tumor protection. To determine whether skin CXCR6$^{-/-}$CD8 $T_{RM}$ cells were capable of providing tumor protection

independently from the lymphoid memory compartment, MAV was induced as previously described and then mice were treated with FTY720 beginning 1 wk prior to tumor re-challenge (Fig 1A). 40 d post-surgery mice were inoculated intradermally with B16F10 melanoma cells on contralateral flanks. In contrast to MAV-affected WT mice which successfully controlled tumor growth, tumors in MAV-affected CXCR6$^{-/-}$ mice had similar growth kinetics to those of unaffected or naïve mice (Fig 6H). Altogether, these results identify CXCR6 as a key component in maintaining long lived CD8 $T_{RM}$ cells in skin.

## Discussion

CD8 $T_{RM}$ cells seed peripheral tissues to establish residence and provide long lasting protection against secondary exposure to their cognate antigen. Since their discovery over a decade ago, we have gained a deeper understanding of their unique functional importance, defined markers for their identification, characterized their tissue-specific residence and identified transcriptional regulators of their identity (1). However, whereas the permanent residence of $T_{RM}$ cells is well appreciated, we have yet to fully decipher the cellular and molecular requirements for their long-term tissue retention, especially in the context of tumor immunity (12, 48). In this study, we describe perifollicular and interfollicular clustering of mouse and human skin CD8 $T_{RM}$ cells in response to melanoma, respectively, and show that persistent clustering with DCs is required for skin $T_{RM}$ cell maintenance and dependent on the interaction between CXCR6 and CXCL16.

Whereas CD11c$^+$ cells have not previously been characterized in $T_{RM}$ cell maintenance, myeloid cells can direct aggregation of effector T cells in response to bacterial and viral pathogens. HSV infection results in perifollicular clustering of leukocytes in both the skin and the female reproductive tract (37). However, the cellular composition of leukocyte clusters varies by location with effector T cells clustering with different APC populations in response to inflammatory chemokines secreted by T cells and/or APCs (37, 38, 39, 40). For example, both CD8 T cells and a CD11b$^+$-monocyte derived DC population produce CCL5 to coordinate perifollicular clustering in the skin, whereas CD11b$^+$ macrophages produce CCL5 to attract T cells in the female reproductive tract (37, 40). Whether T cells and APCs continue to produce chemokines like CCL5 and interact after antigen clearance and during establishment of $T_{RM}$ cell development remains unclear. During anti-tumor responses, CD8 effector T cells readily infiltrate B16F10 tumors in a CXCR3-dependent manner in response to CXCL9 and/or CXCL10 expressed by both tumor cells and CD11c$^+$ cells (48, 49). However, the skin $T_{RM}$ cells that develop after B16F10 removal are dependent on CXCR6 and not CXCR3, indicating that separate pathways direct effector and $T_{RM}$ cell activity. Whether or not the same CD11c$^+$ cells that participate in effector responses switches to express CXCL16 or new CD11c$^+$ cells are recruited to interact with skin $T_{RM}$ cells remains to be determined. Multiple cutaneous DC subsets populate normal skin, including Langerhans cells and conventional DC1 (cDC1) and cDC2 dermal DCs (48, 50). Transcriptional profiling of the two CD11c$^+$ subsets found in MAV-affected skin suggest that the CD11c$^+$CD11b$^{neg}$

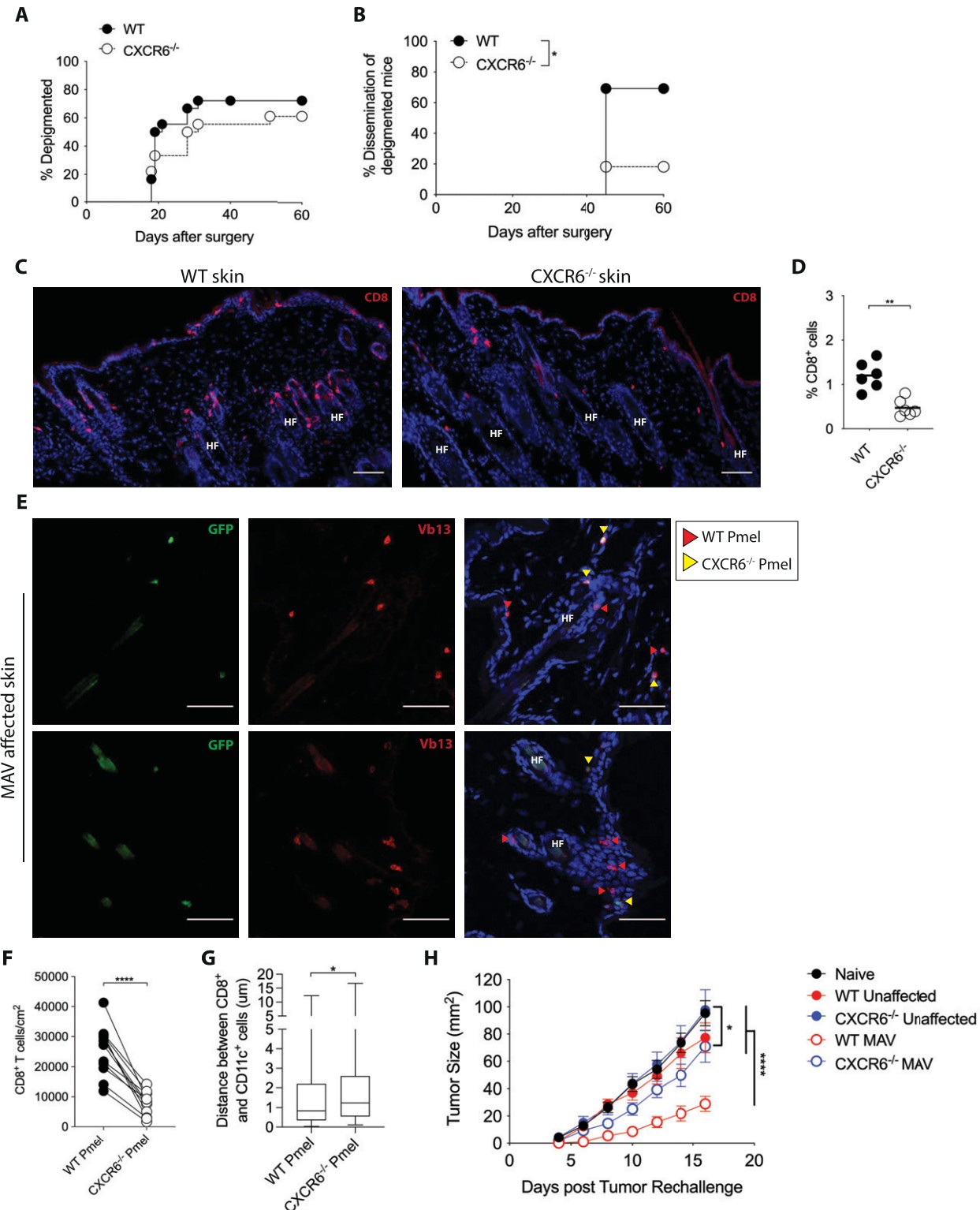

**Figure 6. Expression of CXCR6 on CD8 tissue-resident memory cells is required for their long-term maintenance in skin.**
**(A, B, C, D)** Melanoma-associated vitiligo (MAV) was induced in WT and CXCR6$^{-/-}$ mice as described in Fig 1A. Data are pooled from two independent experiments each with n = 9 mice per group. **(A, B)** Percent of depigmented mice (from [A]) with localized versus disseminated MAV; disseminated MAV occurs when depigmentation has extended 2 cm$^2$ beyond the surgical site. **(C)** MAV skin was analyzed 60 d after surgery from WT and CXCR6$^{-/-}$ mice to detect CD8$\beta$ (red) positive cells; scale bar 25 $\mu$m. **(D)** Percent of CD8 T cells isolated from skin of MAV-affected WT or CXCR6$^{-/-}$ mice. **(E, F, G)** WT recipient mice received 50,000 CD8 T cells at 1:1 WT-Pmel:CXCR6$^{-/-}$ Pmel ratio 1 d before B16F10 tumor injections, mice were then treated as in Fig 1A, skin was harvested 40 d after surgery. **(E)** Localization of WT and CXCR6$^{-/-}$ CD8 Pmel T cells in

cells resemble cDC1s, due to an increase in *XCR1*, *ITGAE*, and *CLEC9A* transcripts, whereas the transcriptional profile of the CD11c⁺CD11b⁺ cells included *SIRPA*, *CD207*, and epithelial cell adhesion molecule (*EPCAM*), indicating this is likely a mixed population of cells comprised of Langerhans cells and cCD2 cells. Future studies that combine spatial and transcriptional data will be informative in identifying the specific myeloid cells that co-aggregate with and are required for skin CD8 $T_{RM}$ cell maintenance.

The functional significance of CD8 $T_{RM}$ cell clustering with CD11c⁺ cell also remains to be determined. Similar to other memory T-cell subsets, CD8 $T_{RM}$ cells require homeostatic cytokines for their long-term survival. In the skin, hair follicle derived IL-15 can support $T_{RM}$ cell survival (51). However, IL-15 must bind to IL-15Rα on neighboring cells and be trans-presented to CD8 T cells in a contact-dependent manner (52). Macrophages and DCs express IL-15Rα (53, 54); therefore, it is plausible that the interacting CD11c⁺ cells promote T cell homeostasis by acting as a source for IL-15/IL-15Rα complexes. Our study demonstrates a requirement for CXCL16-expressing CD11c⁺ cells in maintaining the perifollicular clusters. The interaction of CXCR6 and CXCL16 could be stabilizing cell–cell adhesion between the CD8 $T_{RM}$ cells and CD11c⁺ cells to enhance IL-15 signaling. Alternatively, CXCL16 secreted by CD11c⁺ cells may continuously recruit CD8 $T_{RM}$ cells to the clusters and areas of vitiligo through CXCR6.

CD8 $T_{RM}$ cells are primed to secrete large quantities of IFNγ and TNFα following antigen re-exposure in the context of both viral and tumor responses as well as models of autoimmune vitiligo (4, 12, 55). IFNγ and TNFα have pleiotropic effects including induction of chemokines including CXCL16 by CD11c⁺ cells and keratinocytes (45). Whether or not CD8 $T_{RM}$ cells secrete cytokines to maintain CXCL16 expression by adjacent CD11c⁺ cell remains to be determined; however, the localization of CD8 $T_{RM}$ cell aggregates within depigmented skin argues against the availability of shared melanoma and melanocyte antigens required for stimulating cytokine secretion. Although further studies are needed to conclusively ascertain whether continual antigen sensing is required for CD8 $T_{RM}$ cell maintenance and CXCL16 expression, our data are consistent with viral models in which CD8 $T_{RM}$ cells persist in the absence of antigen (56).

In the absence of antigen, we considered a role for skin microbiota in CD8 $T_{RM}$ cell maintenance. Alterations in the skin microbiome can accelerate the progression of T cell–mediated autoimmune skin diseases including psoriasis, atopic dermatitis, and vitiligo (57, 58, 59). Indeed, decrease in bacterial diversity has been observed in lesional but not nonlesional skin from the same autoimmune vitiligo patient (59), indicating that microbial changes can alter T cell activity in the skin. It is interesting to note that in our mouse model, MAV initially develops at the site of tumor excision and other sites with minor skin abrasions (e.g., ear tag), reminiscent of Koebner phenomenon in vitiligo (60, 61, 62); however, a mechanistic understanding of this effect remains unclear.

Altogether, our findings define an active role for CD11c⁺ cells in not only constraining $T_{RM}$ cells within local tissue but also suggest that they form a supportive niche for long-term $T_{RM}$ cell maintenance at primary tumor sites. To date, therapeutic strategies to increase $T_{RM}$ cells by seeding tissues through "prime and pull" do not yet consider accessory cells that "keep" $T_{RM}$ cells within peripheral tissues for prolonging tumor immunity and progression free survival. It will be important to determine whether the requirement for CD11c⁺ cells to maintain $T_{RM}$ cells within the skin is universally applicable to all $T_{RM}$ cells in response to different tissue-derived tumors and pathogens.

# Materials and Methods

### Human tissue samples

Human samples were collected under the preapproved protocol by the Committee for the Protection of Human Subjects at Dartmouth-Hitchcock Medical Center (DHMC) (no. 00029821) and performed in accordance with ethical guidelines and regulation. De-identified skin biopsy samples from patients with progressing MAV were obtained from the Department of Surgical Oncology at DHMC. Melanoma patients developed vitiligo before or during immunotherapy with nivolumab and/or ipilimumab. Healthy, control, skin was obtained from the DHMC Pathology core.

### Mice

C57BL/6 and CD45.1 mice were purchased from Charles River Breeding Laboratories. *CXCR6⁻/⁻* (stock #:005693), CD11c.DTR (stock #:004509), and Thy1.1⁺Pmel (stock #:005023) mice were purchased from The Jackson Laboratory and bred in-house. To generate CXCR6⁻/⁻Thy1.1⁺Pmel mice, Thy1.1⁺Pmel were crossed with mice CXCR6⁻/⁻ mice. All studies were performed in accordance with the Institutional Animal Care and Use Committee (IACUC) Guidelines at Dartmouth College. Animals were housed within the specific pathogen-free section of the Center for Comparative Medicine and Research at Dartmouth College.

### Study design

All mouse studies were performed in accordance with the IACUC Guidelines at Dartmouth College. Animals were housed within the specific pathogen-free section of the Center for Comparative Medicine and Research at Dartmouth College in standard cages containing a maximum of five mice per cage. Mice had ad libitum access to water and food. In vivo studies used a minimum of five

---

skin from MAV-affected mice. GFP (green), Vβ13 (red), nuclei (blue); scale bar 50 μm. Red arrow heads indicate WT CD8 Pmel T cells, yellow arrow heads indicate CXCR6⁻/⁻ CD8 Pmel T cells. **(F)** Number of WT CD8 Pmel T cells compared with CXCR6⁻/⁻ CD8 Pmel T cells in MAV-skin. **(G)** Distance between a WT CD8 Pmel T-cell or CXCR6⁻/⁻ CD8 Pmel T-cell to CD11c⁺ cells in MAV-affected skin. **(H)** MAV was induced in WT and CXCR6⁻/⁻ mice as described in (A), treatment with FTY720 began 1 wk before tumor re-challenge; on day 40 post-surgery mice were re-challenged with B16F10 melanoma cells on contralateral flanks from the surgical site, tumors were measure on alternating days. **(D, F)** Symbols represent individual mice (D, F); horizontal lines indicate mean. **(A, B, D, E, F, G, H)** n = 18; compiled from two independent experiments (A, B), n = 5–12; representative of two independent experiments (D, E, F, G, H) either through Pmel transfer or transfer of T-cell receptor Pmel retrogenic T cells; Significance was determined by survival curve comparison; *P ≤ 0.02 (B), Mann–Whitney test; **P ≤ 0.005 (D), *P ≤ 0.03 (G), paired *t* test; ****P ≤ 0.0001 (F), two-way ANOVA; ****P ≤ 0.0001 (H).

mice per group. Sample size was altered after experiments were initiated only if tumors re-grew or if the surgical site failed to heal. As indicated, the final data were pooled from identical experiments. All experiments were conducted with fixed end points and were performed in either duplicate or triplicate.

## Generation of bone marrow chimeras

7-wk-old CD45.1 mice received 350 rad per day of γ irradiation from a $^{131}$Cs irradiator on three consecutive days before i.v. injection of CD11c.DTR BM cells (one donor:one recipient). Chimeric mice were used 6 wk post reconstitution.

## Tumor cell lines

To ensure reproducible growth, the B16F10 (B16) mouse melanoma cell line was passaged intradermally (*i.d.*) in C57BL/6 mice. Before using the cell line for experiments, it was tested by the Infectious Microbe PCR Amplification Test and authenticated by the Research Animal Diagnostic Laboratory (RADIL) at the University of Missouri. Tumor cells were cultured in RPMI-1640 containing 10% of heat-inactivated FBS in a humidified incubator at 37°C with 7% $CO_2$.

## Induction of melanoma-associated vitiligo

Mice were inoculated *i.d.* on the right flank with 2 × 10$^5$ B16 cells on day 0. On days 4 and 10 mice were treated with anti-CD4 mAb (clone GK1.5) 250 μg i.p. and tumors were surgically excised on day 12 (30). To track antigen-specific CD8 T-cell responses, mice were adoptively transferred with Pmel cells either 1 d before or 7 d post B16 inoculation. Mice were monitored after surgery for growth of white hairs on the right flank ("localized vitiligo") or beyond ("disseminated vitiligo").

## T cell purification

CD8 T cells were isolated from pooled LNs and spleen of naïve C57BL/6, *CXCR6$^{−/−}$*Thy1.1$^+$Pmel, or WT Thy1.1$^+$Pmel. Negative selection was performed using EasySep Mouse CD8 T Cell Isolation Kit (#19853; Stemcell Tech). Purified naïve T cells were adoptively transferred into recipients or activated before retroviral transduction. CD8 T cells were activated in a 24-well plate at a density of 1 × 10$^6$ per well in complete T-cell medium (RPMI-1640 containing 10% of heat-inactivated FBS, 1% penicillin/streptomycin, 1% non-essential amino acids, and 0.1% β-mercaptoethanol) with Dynabeads Mouse T-Activator CD3/CD28 (#11456D; Thermo Fisher Scientific) and 50 ng/ml recombinant mouse IL-2 (#575406; BioLegend). T cell cultures were passaged with fresh complete T-cell medium and cytokines as needed.

## Plasmids and constructs

The Pmel TCR α and β chain sequences (genbank accession numbers EF154513 and EF154514) separated by P2A self-cleaving sequence were synthesized as a genomic block (gblock) from IDT and cloned into pCMV2.1 MSCV-based retrovirus upstream of an internal ribosomal entry sequence and GFP. Pmel gblock sequence

(TCR α and β sequences underlined, whereas P2A is bolded): (5′-gccggaattcagatctacc<u>atgaaatccttgagtgtttcactagtggtcctgtggctccagttta attgggtgagaagccagcagaaggtgcagcagagcccagaatccctcactgtctcagaggga gccatggcctctctcaactgcactttcagtgatcgttcttctgacaacttcaggtggtacagac agcattctgggaaaggccttgaggtgctggtgtccatcttctctgatggtgaaaaggaagaa ggcagttttacagctcacctcaatagagccagcctgcatgttttcctacacatcagagagccg caacccagtgactctgctctctacctctgtgcagtgaacacaggaaactacaaatacgtcttt ggagcaggtaccagactgaaggttatagcacacatccagaacccagaacctgctgtgtacc agttaaaagatcctcggtctcaggacagcaccctctgcctgttcaccgactttgactcccaaa tcaatgtgccgaaaaccatggaatctggaacgttcatcactgacaaaactgtgctgga catgaaagctatggattccaagagcaatggggccattgcctggagcaaccagacaagc ttcacctgccaagatatcttcaaagagaccaacgccacctaccccagttcagacgttccc tgtgatgccacgttgactgagaaaagctttgaaacagatatgaacctaaactttcaaaacc tgtcagttatgggactccgaatcctcctgctgaaagtagccgggatttaacctgctcatgacg ctgcggctgtggtccagcggctccggagccacgaacttctctctgttaaagcaagcaggagacg tggaagaaaaccccggtcca<b>atgggcaccaggcttcttggctgggcagtgggccccgcggct tcttggctgggcagtgttctgtctccttgacacagtactgtctgaagctggagtcacccagtctcc cagatatgcagtcctacaggaagggcaagctgtttccttttggtgtgaccctatttctgga catgatacccttttactggtatcagcagcccagagaccaggggccccagcttctagtttactttcgg gatgaggctgttatagataattcacagttgccctcggatcgattttctgctgtgaggcctaaaggaa ctaactccactctcaagatccagtctgcaaagcagggcgacacagccacctatctctgtgcca gcagtttccacagggactataattcgcccctctactttgcggcaggcacccggctcactgtgac agaggatctgagaaatgtgactccacccaaggtctccttgtttgagccatcaaaagcagaga ttgcaaacaaacaaaaggctaccctcgtgtgcttggccaggggcttcttccctgaccacgtgg agctgagctggtgggtgaatggcaaggaggtccacagtggggtcagcacggaccctcaggcctac aaggagagcaattatagctactgcctgagcagccgcctgagggtctctgctaccttctggcacaatcc tcgcaaccacttccgctgccaagtgcagttccatgggctttcagaggaggacaagtggccagagggctc acccaaacctgtcacacagaacatcagtgcagaggcctggggccgagcagactgtgggatt acctcagcatcctatcaacaaggggtcttgtctgccaccatcctctatgagatcctgctagggaaa gccaccctgtatgctgtgcttgtcagtacactggtggtgatggctatggtcaaaagaaagaattca</u>tgactcgagtgtttaaacgtcgacggtatcgataagcttcgggatc-3′).

## 293T transfection for viral packaging

293T cells were plated at a density of 2.5 × 10$^6$ cells per 10 cm tissue culture treated dish 1 d before transfection. The following day, cells were transfected by polyethylenimine (PEI) with a luciferase reporter gene using an MSCV-based retrovirus or TCRα/TCRβ specific for gp100 (described above). At 6 h post-transduction, the medium was replaced with 6 ml of complete T-cell medium. At a total of 48 h post transfection, the medium containing virus was collected and passed through a 0.45 μm filter.

## Retroviral transduction

24 h after activation, CXCR6$^{−/−}$Thy1.1$^+$Pmel and Thy1.1$^+$Pmel CD8 T cells were transduced with a PpyRE9 retroviral supernatant to generate Luc$^+$Pmel CD8 T cells. CD8 T cells from WT mice were first transduced with Pmel retroviral supernatant containing 8 μg/ml polybrene and 50 ng/ml recombinant mouse IL-2 then centrifuged at 1,500*g* for 1.5 h, followed by an additional transduction with PpyRE9 retroviral supernatant. In all cases, CD8 T cells were rested at 37°C for 5 h before replacing the retroviral supernatant with fresh complete T-cell medium containing 50 ng/ml recombinant mouse IL-2. Luc$^+$Pmel CD8 T cells were selected with 400 μg/ml G418 for 48 h before adoptive transfer. WT mice transduced with Pmel and PpyRE9 were FACs sorted based on expression of GFP and anti-

mouse Vb13-PE (#140703; BioLegend) and cultured for an additional 24 h in complete T-cell medium containing 50 ng/ml recombinant mouse IL-2 before adoptive transfer into recipient mice at a dose of $2.5-5 \times 10^4$ cells/mouse.

## Imaging of bioluminescent CD8 T cells

Mice were first shaved then injected i.p. with 200 $\mu$l of 15 mg/ml of the luciferase substrate, d-luciferin (#LUCK; Goldbio) in PBS and imaged after 8 min with a Xenogen IVIS-200 system (PerkinElmer). Photon emission was detected with acquisition times ranging from 5 s to 3 min. Analysis of the images was performed using Living Image software (PerkinElmer) by obtaining average radiance per second per cm$^2$ of specified regions of interest.

## CD11c+ cell depletion

For short-term depletion of CD11c expressing cells, CD11c.DTR mice received 3–50 ng *i.d.* injections of DT (#D0564; Sigma-Aldrich) at the site of depigmentation over the course of 1 wk. In experiments where CD11c$^+$ cell depletion was continuously maintained, CD11c.DTR bone marrow chimeric mice received 0.25 $\mu$g of DT *i.p.* every 3 d beginning on day 30 post-surgery and lasting until termination of the experiment.

## Flow cytometry

On indicated days after tumor excision, mice were euthanized and inguinal (tumor-draining) lymph nodes, spleen, and a 2-cm$^2$ patch of skin overlapping the surgery site as well as distal skin were harvested. Lymphoid tissues were mechanically dissociated. Skin was minced and incubated in 2 mg/ml Collagenase Type IV (Worthington Biochemical Corporation), 0.2 mg/ml DNase I (Sigma-Aldrich), and 2% FBS in HBSS at 37°C for 25 min with a magnetic stir bar. Remaining skin fragments were mechanically dissociated in RPMI-1640 containing 10% FBS and 2 mM EDTA. Cell suspensions were first stained for live cells with Zombie Aqua Fixable Viability dye (#423101; BioLegend) then Fc receptors were blocked using anti-CD16 and anti-CD32 antibodies (#BE0307; Bio X Cell). Samples were stained for 30 min on ice with various antibody combinations. Antibodies from BioLegend: anti-mouse CD45-APC/Fire 750 (clone 30-F11; #147713), anti-mouse CD8α-PerCP/Cynine5.5, -PE/Cy7 (clone 53-6.7; #100733, #100721), anti-mouse Thy1.1-PerCP/Cynine5.5 (clone OX-7; #202515), anti-mouse CD103-Alexa Fluor 647, -FITC (clone 2E7; #121409, #121419), anti-mouse CD44-APC/Fire 750 (clone IM7; #103061), anti-mouse CD62L-Brilliant Violet 510 (clone MEL-14; 104441), anti-mouse CXCR6-Brilliant Violet 421 (clone SA051D1; #151109), anti-mouse CD11c Alex Fluor 647 (clone N418; #117314), anti-mouse CD11b-Alexa Fluor 488 (clone M1/70; #101219); Bioss: anti-mouse CXCL16 Alexa Fluor 488 (polyclonal; #bs-1441R-A488). For intracellular staining of CXCR6, cells were fixed/permeabilized using reagents from the Foxp3 Staining Kit (#421403; BioLegend) following the manufacturer's protocol. Flow cytometry was performed on a MACSQuant 10 Analyzer (Miltenyi), and data were analyzed using FlowJo software (Tree Star).

## Immunohistochemistry (IHC)

For histological examination, tissues were fixed with 10% formalin in phosphate-buffered saline, embedded in paraffin, and cut into 4 $\mu$m sections. The following primary antibodies were used to stain the paraffin embedded sections: anti-human CD8 (clone 4B11; #CD8-4B11-L-CE-H; Leica Microsystem) anti-human CD11c (clone EP1347Y; #ab52632; Abcam), anti-human CXCL16 (clone GT516; #MA5-27845; Thermo Fisher Scientific), anti-human CXCR6 (polyclonal; #PA5-33462; Thermo Fisher Scientific), and Fontana Masson to detect melanin. IHC stains were performed using Leica Bond Max and RX Automated stainer (Leica Microsystems), Bond Epitope Retrieval 2 (Leica Microsystem), and ChromoPlex 1 Dual Detection for BOND kit (Leica Microsystem) all according to the manufacturers' instructions. Immunohistochemical stains of human tissues were conducted by the Dartmouth Pathology Shared Resources.

## Immunofluorescence (IF)

Excised skins were incubated in 4% PFA for 15 min at 4°C followed by a 1-h incubation in 30% sucrose all in PBS. Skins were embedded in optimum cutting temperature (Tissue Tek; Sakura). 10 $\mu$m sections were cut using a cryostat, air-dried, fixed in cold methanol, and then rehydrated in PBS. Sections were blocked with 5% BSA, 1% goat serum, 1% rat serum and 1% donkey serum in PBS for 1 h at RT and then stained overnight at 4°C with combinations of the following directly conjugated antibodies from BioLegend: anti-mouse CD8$\beta$-Alexa Fluor 555 (clone 53-6.7), anti-mouse CD11c Alexa Fluor 647 (clone N418), and anti-mouse CD11b Alexa Fluor 488 (clone M1/70; #101219); Bioss: anti-mouse CXCL16 Alexa Fluor 488 (polyclonal; #bs-1441R-A488); Hoechst 33324 (#H3570; Thermo Fisher Scientific). Slides were mounted using ProLong Diamond Antifade Reagent (P36961; Thermo Fisher Scientific) and left overnight at room temperature to set. To detect GFP, skin was rehydrated in PBS, permeabilized with 0.25% Triton X-100 in PBST (PBS and 1% Tween 20) for 10 min at RT, then blocked with 5% BSA and 0.1% Triton X-100 in PBS at RT for 1 h. Anti-mouse GFP-Alexa Fluor 555 (#A-31851; Thermo Fisher Scientific) was diluted in 1% BSA with 0.1% Triton X-100 and incubated ON at 4°C.

## CD8$\beta$ conjugation to Alexa Fluor 555

Purified anti-mouse CD8$\beta$ (clone 53-5.8; #140402; BioLegend) was conjugated to Alexa Fluor 555 using the Alexa Fluor 555 Antibody Labeling Kit (#A20187; Thermo Fisher Scientific) following the manufacturer's protocol.

## Image acquisition

Images were acquired with a Zeiss LSM 800 microscope fitted with GaAsP detectors, using a 40× Plan-Apochromat 1.4NA objective. Patient skin and whole murine skin scans were acquired with the PerkinElmer Vectra three automated Olympus upright BX51 fluorescence microscope using the 10× UPlan SApo NA 0.40 WD and the 20× UPlan SApo NA 0.75 WD objectives.

## IHC image processing and analysis

Number of cells/cm$^2$ and distances between cells in patient skin was manually quantified/measured using Fiji (ImageJ). inForm image analysis software (PerkinElmer) was used for colocalization

measurements. Signal crosstalk was eliminated by first creating a spectral library to define the spectral curve for each chromogen (Fast Red, Vina Green, and hematoxylin). The spectral library was then used to unmix the signals on the multicolored slides by recognizing the unique spectral curves. After spectral unmixing, the pixel-based colocalization image analysis of the inForm software package was used to determine the percentage of cells co-expressing target markers.

### IF image processing and analysis

Images of murine skin were analyzed and processed with Fiji (ImageJ) and Imaris 9.5 software (Bitplane). To determine the shortest distance between two CD8 T cells, ND plugin for ImageJ was used according to the developer's instructions (63). Images were first converted to 8-bit, image was thresholded to highlight features of interest, then the built in "Analyze particle" function was run to indicate size and circularity of cells, followed by running the ND plugin. To determine mean fluorescence of CXCL16 surrounding hair follicles, an outline was drawn around randomly selected hair follicles and mean fluorescence measured, along with several adjacent background readings. The total corrected cellular fluorescence calculated (total corrected cellular fluorescence = integrated density–[area of selected cell × mean fluorescence of background readings]) (64). Imaris 9.5 software was used to determine the shortest distance between a CD8 T cell and the closest CD11c$^+$ cell. The epidermis was first masked, CD8 T cells were identified using the spots tool, whereas CD11c$^+$ cells were identified using the surface tool. After identification of both cell types Imaris' shortest distance tool was used to calculate the distances between the two.

### 3′ RNA sequencing

For RNA-sequencing CD8 T cells, CD11c$^+$CD11b$^{neg}$ and CD11c$^+$CD11b$^+$ monocytes were sorted from MAV skin 50 d after surgical removal of B16F10 tumor. Each tissue was digested as described above, two mice were pooled before staining, for a total of three experimental samples. Single-cell suspensions were prepared for each tissue and stained with anti-mouse CD8a-PerCP/Cynine5.5 (clone 53-6.7; #100773; BioLegend), anti-mouse CD11c Alexa Fluor 647 (clone N418; #117314; BioLegend), anti-mouse CD11b Alexa Fluor 488 (clone M1/70; #101219; BioLegend), and anti-mouse CD45.2 Alexa Fluor 488 (clone 104; #109821; BioLegend). The various cell populations were sorted directly into 200 $\mu$l QIAGEN RLT buffer using an ARIA-II cell sorter (BD Biosciences). Total RNA was purified using RNeasy Mini Kit (#74134/74136; QIAGEN) following the manufacturer's protocol. From the extracted RNA, cDNA was made using the SMART-Seq v4 Ultra Low Input Kit (Takara) and 10 cycles of cDNA amplification. Libraries were generated from 10 ng cDNA using the Nextera DNA Flex library prep kit (Illumina). Libraries underwent quality control by Fragment Analyzer (Agilent) and Qubit (Thermo Fisher Scientific) to determine the size distribution and the quantity of the libraries. Libraries were sequenced on a NextSeq 500 (Illumina). Fastq files were aligned to the mm10 genome using bowtie2 (65) and normalized to obtain Transcripts Per Kilobase Million (TPM) for each RNA-seq sample using the software RSEM (66, 67).

### 3′ RNA-sequencing analysis

Differentially expressed genes were determined using R package DESeq2 (68), $P$-values < 0.05 were determined to be significant. The top 100 genes along with myeloid-specific genes and skin T$_{RM}$–specific genes were chosen from each population, these genes with their corresponding TPM were used to generate heat maps using the browser-based software, Morpheus (https://software.broadinstitute.org/morpheus).

### Statistical analyses

When comparing statistical differences between two groups an unpaired two-tailed $t$ test was used, when comparing three distinct groups one-way ANOVA was used. All statistical analyses were performed in Prism 8 software (GraphPad) and data were considered significant if $P \leq 0.05$.

## Data Availability

Bulk RNA-seq data generated can be found at Gene Expression Omnibus under accession code GSE180647.

## Supplementary Information

## Acknowledgements

We thank G Ward in the DartLab FACS core facility for cell sorting; E DuFour in the Dartmouth CCMR for mouse handling; P Zhang in DartLab for performing tumor resections; R O'Meara in the Dartmouth-Hitchcock Medical Center Pathology core for immunostaining human skin sections; FW Kolling IV for RNA sequencing; and members of the Huang and Turk laboratories for discussions and technical assistance. This work was supported by the National Institutes of Health grants R01-AI089805 to YH Huang, R01-CA225028 to MJ Turk, T32-AI007363 to JL Vella, the Burroughs Wellcome Fund, Big Data in the Life Sciences Training Program to HE Lust, Air Force Office of Scientific Research grant FA9550-18-1-0017 to BR Branchini, and P30-CA023108 to Dr S Leach which supports the Norris Cotton Cancer Center's genomics, microscopy, and flow cytometry cores.

### Author Contributions

JL Vella: conceptualization, data curation, formal analysis, validation, investigation, visualization, methodology, and writing—original draft, review, and editing.
A Molodtsov: data curation, formal analysis, and writing—review and editing.
CV Angeles: data curation, investigation, and writing—review and editing.
BR Branchini: methodology and writing—review and editing.
MJ Turk: conceptualization, supervision, funding acquisition, and writing—original draft, review, and editing.

YH Huang: conceptualization, data curation, supervision, funding acquisition, investigation, project administration, and writing—original draft, review, and editing.

## Conflict of Interest Statement

The authors declare that they have no conflict of interest.

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
