## [Reviewer comments · Life Science Alliance]

Life Science Alliance

Dendritic cells maintain anti-tumor immunity by positioning CD8 skin resident memory T cells

Jennifer Vella, Aleksey Molodtsov, Christina Angeles, Bruce Branchini, Mary Turk, and Yina Huang
DOI: <https://doi.org/10.26508/lsa.202101056>

Corresponding author(s): Yina Huang, Geisel School of Medicine

Review Timeline:	Submission Date:	2021-02-16
	Editorial Decision:	2021-03-27
	Revision Received:	2021-06-19
	Editorial Decision:	2021-07-14
	Revision Received:	2021-07-22
	Accepted:	2021-07-23

Transaction Report:

March 27, 2021

Re: Life Science Alliance manuscript #LSA-2021-01056-T

Dr. Yina Hsing Huang
Geisel School of Medicine
Pathology, Microbiology and Immunology
1 Medical Center Dr
Lebanon, NH 03756

Dear Dr. Huang,

Thank you for submitting your manuscript entitled "Dendritic cells maintain anti-tumor immunity by positioning CD8 skin resident memory T cells" to Life Science Alliance. The manuscript was assessed by expert reviewers, whose comments are appended to this letter. We would like to invite you to submit a revised version of this manuscript that addresses all of the reviewers' concerns.

We apologize for this unusual and extended delay in getting back to you. As you will note from the reviewers' comments below, the reviewers are interested in these findings, but have raised a number of significant questions and concerns, all of which must be addressed prior to further consideration of the manuscript at LSA.

Thank you for this interesting contribution to Life Science Alliance. We are looking forward to receiving your revised manuscript.

Sincerely,

Shachi Bhatt, Ph.D.

Executive Editor

Life Science Alliance

<https://www.lsjournal.org/>

Interested in an editorial career? EMBO Solutions is hiring a Scientific Editor to join the international Life Science Alliance team. Find out more here -

https://www.embo.org/documents/jobs/Vacancy_Notice_Scientific_editor_LSA.pdf

B. MANUSCRIPT ORGANIZATION AND FORMATTING:

Reviewer #1 (Comments to the Authors (Required)):

This study addresses mechanisms that maintain melanoma-specific CD8+ resident memory (T_{rm}) cells in the skin after surgical removal of the tumor. In this well-characterized experimental model, 60 - 80% of wild type mice developed autoimmune melanoma-associated vitelligo (MAV), and CD8+ T cell-mediated protection to tumor challenge. The authors propose that secretion of the

chemokine CXCL16 by CD11c⁺ cells in the skin plays a critical role in retention of protective tumor-specific CD8⁺ T_{RM}. They observed that hair follicles in depigmented (MAV⁺) skin were surrounded by clusters of CD8b⁺ cells and CD11c⁺ cells; similar clusters were seen in skin from human patients with MAV. In mice, the CD11c⁺ cells broadly divided into CD11b⁺ and CD11b⁻ subpopulations. Transcriptional analysis suggested that the CD8b⁺ cells were CD8⁺ resident memory T cells; the CD11c⁺ CD11b⁻ cells were cDC1; and the CD11c⁺ CD11b⁺ cells were a mixed population including skin cDC2. Depletion of CD11c⁺ cells resulted in loss of the tight clusters, while the CD8b⁺ cells spread out into the surrounding skin and subsequently left or died. Chemokine and chemokine receptor expression, together with data from chemokine receptor-deficient mice, suggested that CD11c⁺ cell expression of CXCL16 is critical for retention of CD8⁺ T_{RM} (which express the CXCR6 receptor) within the follicle-associated clusters. This is a really interesting study, with important mechanistic and therapeutic implications.

Major issues:

The very minimal characterization of the CD11c⁺ cell populations is a significant concern. Skin from MAV⁺ and MAV⁻ mice should be analyzed (even if this isn't feasible for sorting) using the universal cDC panel, with additional markers to distinguish skin DC subsets, described by Williams and colleagues in 2016 (*Immunity* 45: 669). (n.b. F4/80 is not solely a macrophage marker - it's also expressed on a skin cDC2 population.) The authors should also describe why the transcript profiles in Fig. S3d are consistent with different cDC subsets (e.g. expression of XCR1 versus SIRPA). The substantial variation in the proportions of CD11c⁺ CD11b⁻ versus CD11c⁺ CD11b⁺ cells, e.g. in Fig. 2h (mostly CD11c⁺ CD11b⁻) compared with Fig. S3b (mostly CD11c⁺ CD11b⁺) should also be discussed.

It would be helpful to know whether differences in CXCR6 expression (Fig. 5c) are due to increased autofluorescence in activated and T_{RM} CD8⁺ T cells compared with naïve cells. The authors should also mention that the adoptively transferred Luc⁺Pmel CD8⁺ T cells (unlike endogenous cells) were primed in the absence of CD4⁺ T cells (though this probably doesn't matter since the priming cDC received help earlier).

In several instances, the way the data are presented really minimizes their impact. Maybe some axis labels are also incorrect? For instance, the images in Fig. 1b show convincingly that CD8b⁺ cells (largely CD103⁺ KLRG1⁻, Fig. S1) cluster round hair follicles in MAV⁺ but not MAV⁻ mice. Fig. 1c shows the average distance between CD8b⁺ cells; it's really hard to see how this agrees with the data in Fig. 1b. Fig. 1d suggests that there are {less than or equal to}0.016 clusters/cm², again, hard to reconcile with Fig. 1b, where there are two clusters very close together. Fig. S5b shows that there are 1 to 4 CD8⁺ T cells per cm², again inconsistent with the images in Fig. S5a.

Minor issues:

- 1: Several of the figures (e.g. Fig 3b) are much too small, and the enlarged PDF is very unclear.
- 2: Should Fig. 4e on p10, line 7, be Fig. 4g?

Reviewer #2 (Comments to the Authors (Required)):

The authors report a role for the interaction between CXCR6 expressed by memory resident CD8⁺T cells (TRM) and CXCL16-expressing dendritic cells in the retention of TRM. This is a compelling work that adds a new mechanism of regulation of CD8 TRM and the importance of the CXCL16-CXCR6 axis in TRM positioning within tissues.

Some suggestions to increase the impact of this work

- In vitro data on the interaction between dendritic cells and CD8 T_{RM} and the role of CXCR6-CXCL16 would strengthen this observation of firm adhesion of T_{RM} and dendritic cells in skin.
- Supplementary Fig 5: Is the number of T_{RM} and not only total CD8⁺T cells affected in mice depleted in DC or with CXCR6 deficient CD8⁺T cells.
- Fig 3 shows membrane CXCL16 expressed by DCs. Is CXCL16 also secreted by these cells and what is the respective expression of the membrane and secreted form.
- Fig 4 E does not concern the decrease of CD8 rather 4G
- The small differences in tumor growth observed with CXCR6 KO mice could be explained by the fact that the B16-F10 tumor is not a highly CD8-dependent lineage
- In the introduction, references 8, 9 refer rather to CD8⁺T cells in general. It would be better to cite for T_{RM}, the articles by Edwards J et al Clin Cancer Res 2018 and Cognac S et al Cell Rep Med 2020) or a review (Mami-Chouaib F J Immunother Cancer 2018). In the discussion, the article by Shimaoka T et al J Leuk Biol 2004 and Karaki S J Immunother Cancer 2021 on the role of CXCR6 in another model could be mentioned

Reviewer #3 (Comments to the Authors (Required)):

In this study, Vella et al characterized the role of CXCR6⁺ T cells and CXCL16⁺ APCs in melanoma-associated vitiligo (MAV). The authors conclude that while CXCR6 is dispensable for CD8 T cell activation, effector cell recruitment and melanocyte destruction, that it does play a role in T_{RM} tethering to CD11c⁺ APC populations in the skin. Generally, this manuscript does fall in the scope of Life Science Alliance, which includes descriptive data and important negative data of value to the community (presumably the authors were hoping that CXCR6 deficiency in Pmel T cells would prevent disease in mice, which it did not). CXCR6 in tumor-infiltrating lymphocytes has not previously been explored in MAV, which is interesting and novel. Therefore, I do think this manuscript should ultimately be published. However, there are many areas of the manuscript that require more rigor/clarification as I will delineate below.

My major issues with this study as presented are that (1) the authors rely on immunohistochemistry of CD8 and CD11c to identify T_{RM} and DC/APC. CD8 in the skin can have many phenotypes, and T_{RM} markers should have been used. CD11c can be expressed by many other immune cell types including T and B cells, especially during active inflammation. (2) The authors interpret their data in Fig 6b to mean that CXCR6 is required for T_{RM} maintenance because depigmentation did not disseminate in CXCR6^{-/-} mice. However, I might interpret this to mean that it is required for establishment of new and/or spreading lesions, in which T_{RM} could form in the new area. (3) The authors do not discuss location of T_{RM} and CD11c⁺ APCs in human skin. Presumably these reside in the interfollicular epidermis, not just in hair follicles, which is a limitation of their mouse model. As with mouse data, the authors are calling these human T_{RM} and APCs, but do not include additional immunohistochemical markers for identification.

My specific points are as follows:

Major:

-Could you please show the changes in CXCL16 and CXCR6 expression (kinetics) during the initiation and establishment of MAV?

-P12 Fig 6b - "depigmentation did not disseminate in CXCR6^{-/-} mice"; please discuss alternate interpretations of these data in results/discussion. Could you please clarify if you counted CD8⁺ T

cells or Trm specifically in CXCR6 KO mice? It would be valuable to present the data on the number of Trm in CXCR6 KO T cell transfer mice over time.

-Was the human healthy control skin age and sex matched with the melanoma patient samples?

-Fig 2D were the Trm in human MAV only in hair follicles, or in the inter-follicular spaces? Please include a discussion point describing the differences between location in human and mouse Trm in MAV and why this might occur (i.e. pigmentation in B6 mouse skin is primarily derived from hair/hair follicles whereas humans have inter-follicular melanocytes).

-Fig 3D please provide orientation/labeling on the histology for the human samples. Are these CXCL16+ APC clusters interfollicular or in hair follicles? Are they proximal to the tumor site or from areas of distal depigmentation? I would be interested to know if these are in distal depigmented areas, which would mirror the spread of depigmentation in the mice, or if they are limited to proximal sites.

-Fig 4G & 5F how many of these CD8 T cells express TRM phenotypic markers?

-Fig 6B & C, the authors use these data to support their conclusion that CXCR6 is required for Trm maintenance. However, the data could be interpreted as CXCR6/CXCL16 is required for lesion spread (i.e. establishment of new lesions and formation of new Trm). This is a major difference in my interpretation of the authors' data versus how it is presented in this manuscript.

-Fig S1C compares Pmel to endogenous CD8 T cells and concludes that differences represent Trm vs other T cell phenotypes. However, the authors present data in Fig 1 (and in previous publications from Turk lab) showing MAV without Pmel transfer, so presumably some of these endogenous CD8 T cells are in fact antigen-specific. It would be interesting to know if the endogenous antigen-specific Trm share similar transcriptional profiles or if the Pmel cells are unique, which might be due to their TCR, the fact that they are transgenic, etc.

-Other type of immune cells including B cells, T cells, and NK cells also express CD11c particularly in inflammatory conditions. Can the authors provide further justification for calling these DCs?

Minor:

-to prove tethering, MP-IVM (in vivo microscopy) could have been conducted. I understand this is a very costly, technically-challenging experiment, so for this study I would not require this type of analysis. The authors do show clustering/distance by static microscopy, but I just want to note here that this is a limitation for interpretation.

-Please clarify that Fig 3a is from mouse (not human)?

-Fig S1 figure legend refers to (D) but not (C) heatmap. Please correct this.

-P11 after ref to (Fig 5d) is a sentence fragment, suggest removing "While"

-P15 discussion the phrase "yet, the underlying factors that trigger their formation and persistence remain unclear." Is redundant w previous sections, suggest to remove

-P16 "tempting to consider" is a colloquialism, suggest rewording

-P16 "It is interesting to note that in our mouse model, MAV initially develops at the site of tumor excision and other sites with minor skin abrasions (e.g. ear tag); however, a mechanistic understanding of this effect remains unclear" This is called koebnerization; both melanoma and vitiligo can exhibit koebner phenomenon. I suggest reading/citing some of the following literature: G Weiss et al JEADV 2002, Sagi & Trau Clinics in Derm 2011, N van Geel et al Clin Lab Invest 2012, Khurram et al Ann Dermatol 2017, Larsabal et al JAAD 2017, R C Liu et al Clin Exp Dermatol 2019, Redondo et al Derm Surgery 2006, Federica et al Dermatologic Surgery 2009,

-P22 "All performed by Dartmouth's Pathology Shared Resource" is a sentence fragment, suggest adding "immunohistochemistry procedures were"

Point-by-point reply to all reviewers' comments

Reviewer 1:

This study addresses mechanisms that maintain melanoma-specific CD8⁺ resident memory (Trm) cells in the skin after surgical removal of the tumor. In this well-characterized experimental model, 60 - 80% of wild type mice developed autoimmune melanoma-associated vitelligo (MAV), and CD8⁺ T cell-mediated protection to tumor challenge. The authors propose that secretion of the chemokine CXCL16 by CD11c⁺ cells in the skin plays a critical role in retention of protective tumor-specific CD8⁺ Trm. They observed that hair follicles in depigmented (MAV⁺) skin were surrounded by clusters of CD8b⁺ cells and CD11c⁺ cells; similar clusters were seen in skin from human patients with MAV. In mice, the CD11c⁺ cells broadly divided into CD11b⁺ and CD11b⁻ subpopulations. Transcriptional analysis suggested that the CD8b⁺ cells were CD8⁺ resident memory T cells; the CD11c⁺ CD11b⁻ cells were cDC1; and the CD11c⁺ CD11b⁺ cells were a mixed population including skin cDC2. Depletion of CD11c⁺ cells resulted in loss of the tight clusters, while the CD8b⁺ cells spread out into the surrounding skin and subsequently left or died. Chemokine and chemokine receptor expression, together with data from chemokine receptor-deficient mice, suggested that CD11c⁺ cell expression of CXCL16 is critical for retention of CD8⁺ Trm (which express the CXCR6 receptor) within the follicle-associated clusters. This is a really interesting study, with important mechanistic and therapeutic implications.

Rev 1, major concern 1: The very minimal characterization of the CD11c⁺ cell populations is a significant concern. Skin from MAV⁺ and MAV⁻ mice should be analyzed (even if this isn't feasible for sorting) using the universal cDC panel, with additional markers to distinguish skin DC subsets, described by Williams and colleagues in 2016 (*Immunity* 45: 669). (n.b. F4/80 is not solely a macrophage marker - it's also expressed on a skin cDC2 population.) The authors should also describe why the transcript profiles in Fig. S3d are consistent with different cDC subsets (e.g. expression of XCR1 versus SIRPA). The substantial variation in the proportions of CD11c⁺ CD11b⁻ versus CD11c⁺ CD11b⁺ cells, e.g. in Fig. 2h (mostly CD11c⁺ CD11b⁻) compared with Fig. S3b (mostly CD11c⁺ CD11b⁺) should also be discussed.

Response: At Rev 1's suggestion, we conducted a detailed characterization of skin CD11c⁺ cells after consulting with our colleague, Dr. Claudia Jakubzick, a myeloid cell expert. Using CD11c, MHC II, CD64, XCR1 and EpCAM markers, we were able to distinguish skin macrophages from cDCs and Langerhan cells (LCs) (see below).

We found no difference in total numbers of macrophages, cDC1s and LCs in unaffected and MAV affected skin. However, it is important to note that the skin is populated with many CD11c⁺ cells even at steady state. It is currently unclear to us whether the CD11c⁺ cells that cluster with CD8

T_{RM} cells are recruited from the blood or the surrounding skin. If the latter is true, then an increase in skin CD11c+ cells would not be expected. We included this in the discussion (lines 295-298). Immunohistochemistry allowed us to detect at least 2 populations clustered with CD8 T_{RM} cells based on differential CD11b and CD11c expression (Fig 2h, Sup Fig 4a), indicating that multiple CD11c+ populations are present. In response to Rev 1 comments, we also tried to stain skin sections with antibodies against XCR1 and Sirp1a but were unsuccessful in detecting specific staining. We are actively investigating whether specific myeloid cells are required for CD8 T_{RM} cell maintenance using inducible lineage-specific DTR mice (e.g. XCR1-cre x zDC-loxStop lox-DTR, CD207-DTReGFP mice). However, we believe these experiments are beyond the scope of this study, which reports the perifollicular clustering of skin CD8 T_{RM} and CD11c+ cells and the requirement for CXCR6 in CD8 T_{RM} cells in the context of MAV-associated tumor protection.

Rev 1, major concern 2: *It would be helpful to know whether differences in CXCR6 expression (Fig. 5c) are due to increased autofluorescence in activated and Trm CD8+ T cells compared with naïve cells. The authors should also mention that the adoptively transferred Luc+Pmel CD8+ T cells (unlike endogenous cells) were primed in the absence of CD4+ T cells (though this probably doesn't matter since the priming cDC received help earlier).*

Response: We drew our CXCR6 gates based on a fluorescence-minus-one (FMO) control sample in which all antibodies except CXCR6 were used to stain cells. In our experience, this is the best method to distinguish true signal from autofluorescence (see below). In addition, the text has been modified to reflect the activation status of Luc+Pmel CD8+ T cells (line 202).

Fluorescence minus one control sample, in which CXCR6 antibody was omitted from the stain in order to allow proper gating.

Rev 1, major concern 3: *In several instances, the way the data are presented really minimizes their impact. Maybe some axis labels are also incorrect? For instance, the images in Fig. 1b show convincingly that CD8b+ cells (largely CD103+ KLRG1-, Fig. S1) cluster round hair follicles in MAV+ but not MAV- mice. Fig. 1c shows the average distance between CD8b+ cells; it's really hard to see how this agrees with the data in Fig. 1b. Fig. 1d suggests that there are {less than or equal to}0.016 clusters/cm2, again, hard to reconcile with Fig. 1b, where there are two clusters very close together. Fig. S5b shows that there are 1 to 4 CD8+ T cells per cm2, again inconsistent with the images in Fig. S5a.*

Response: Thank you for making these points. We have added Fig 1d to depict the number of CD8 T cells/area of skin, and as Rev 1 predicted, the data are much more impactful.

Rev 1: detailed comment 1. *Several of the figures (e.g. Fig 3b) are much too small, and the enlarged PDF is very unclear*

Response: Figures are now larger. Our apologies but limits to file size may still affect the quality of IHC images.

Rev 1: detailed comment 2. *Should Fig. 4e on p10, line 7, be Fig. 4g?*

Response: Yes, this has been corrected.

Reviewer 2:

The authors report a role for the interaction between CXCR6 expressed by memory resident CD8+T cells (TRM) and CXCL16-expressing dendritic cells in the retention of TRM.

This is a compelling work that adds a new mechanism of regulation of CD8 TRM and the importance of the CXCL16-CXCR6 axis in TRM positioning within tissues.

Some suggestions to increase the impact of this work

Rev 2, suggestion 1: *In vitro* data on the interaction between dendritic cells and CD8 TRM and the role of CXCR6-CXCL16 would strengthen this observation of firm adhesion of TRM and dendritic cells in skin.

Response: We agree with this suggestion although we have not been able to conduct additional functional assays with CD8 T_{RM} cells and dendritic cells isolated from skin. This is likely due to the extensive isolation process (i.e. collagenase digestion). We have softened the text in our discussion of CXCR6-CXCL16 in mediating firm adhesion, and instead reference published studies concluding that the properties of CXCR6 (cytoplasmic DRF motif favors adhesion over migration – Koenen *et al.* 2017) and CXCL16 (membrane associated) favor a role in adhesion although a role in migration cannot be excluded (lines 77, 64-68, 315-316).

Rev 2, suggestion 2: *Supplementary Fig 5 (now Supp Fig 6): Is the number of TRM and not only total CD8+T cells affected in mice depleted in DC or with CXCR6 deficient CD8+T cells. (TRM panel for bulk CD8s in MAV skin)*

Response: Yes, skin T_{RM} cells are reduced in CXCR6 deficient mice. In fact, 95% of CD8+ T cells in MAV skin express TRM markers, CD103 and CD69. We have now added Supp. Fig. 1a.

Rev 2, suggestion 3: *Fig 3 shows membrane CXCL16 expressed by DCs. Is CXCL16 also secreted by these cells and what is the respective expression of the membrane and secreted form.*

Response: We were unable to directly measure secreted CXCL16 from isolated skin cells. We have previously used PCR primers that distinguished all CXCL16 isoforms versus transmembrane bound CXCL16; however, there are no primers capable of selectively detecting the secreted protein. We now discuss possible roles for both secreted and surface CXCL16.

Rev 2, suggestion 4: *Fig 4 E does not concern the decrease of CD8 rather 4G*

Response: This has been corrected.

Rev 2, suggestion 5: *The small differences in tumor growth observed with CXCR6 KO mice could be explained by the fact that the B16-F10 tumor is not a highly CD8-dependent lineage*

Response: Indeed, B16-F10 tumors are considerably less immunogenic compared to other tumors although they do grow more robustly in CD8-deficient mice (see figure below in which 2×10^5 B16F10 cells were inoculated into either wt or CD8-deficient mice). We have also reported that wt CD8 T_{RM} cells that develop in the MAV model are necessary and sufficient to mount antitumor memory responses upon B16F10 re-challenge (Malik *et al. Sci Immunol.* 2017).

Rev 2, suggestion 6: *In the introduction, references 8, 9 refer rather to CD8+T cells in general. It would be better to cite for TRM, the articles by Edwards J et al Clin Cancer Res 2018 and Corgnac S et al Cell Rep Med 2020) or a review (Mami-Chouaib F J Immunother Cancer 2018). In the discussion, the article by Shimaoka T et al J Leuk Biol 2004 and Karaki S J Immunother Cancer 2021 on the role of CXCR6 in another model could be mentioned*

Response: We agree and have added references for CD8 T_{RM} cells and CXCR6.

Reviewer 3:

In this study, Vella et al characterized the role of CXCR6+ T cells and CXCL16+ APCs in melanoma-associated vitiligo (MAV). The authors conclude that while CXCR6 is dispensable for CD8 T cell activation, effector cell recruitment and melanocyte destruction, that it does play a role in Trm tethering to CD11c+ APC populations in the skin. Generally, this manuscript does fall in the scope of Life Science Alliance, which includes descriptive data and important negative data of value to the community (presumably the authors were hoping that CXCR6 deficiency in Pmel T cells would prevent disease in mice, which it did not). CXCR6 in tumor-infiltrating lymphocytes has not previously been explored in MAV, which is interesting and novel. Therefore, I do think this manuscript should ultimately be published. However, there are many areas of the manuscript that require more rigor/clarification as I will delineate below.

My major issues with this study as presented are that (1) the authors rely on immunohistochemistry of CD8 and CD11c to identify Trm and DC/APC. CD8 in the skin can have many phenotypes, and Trm markers should have been used. CD11c can be expressed by many other immune cell types including T and B cells, especially during active inflammation. (2) The authors interpret their data in Fig 6b to mean that CXCR6 is required for Trm maintenance because depigmentation did not disseminate in CXCR6^{-/-} mice. However, I might interpret this to mean that it is required for establishment of new and/or spreading lesions, in which Trm could form in the new area. (3) The authors do not discuss location of Trm and CD11c+ APCs in human skin. Presumably these reside in the interfollicular epidermis, not just in hair follicles, which is a limitation of their mouse model. As with mouse data, the authors are calling these human Trm and APCs, but do not include additional immunohistochemical markers for identification. (cite Jichang's paper; call human CD11c+ cells)

My specific points are as follows:

Rev 3, comment 1: *Could you please show the changes in CXCL16 and CXCR6 expression (kinetics) during the initiation and establishment of MAV?*

Response: Data for CXCL16 expression on CD11c+ cells on days 7 and 14 have now been added to Fig. 3b. CXCR6 expression in naïve and on days 7, 14 and 35 are in Fig. 5c, d.

Rev 3, comment 2: *P12 Fig 6b - "depigmentation did not disseminate in CXCR6^{-/-} mice"; please discuss alternate interpretations of these data in results/discussion.*

Response: We agree with Rev 3 and have now added the alternative interpretation that CXCR6 may be required for establishment of new and/or spreading lesions has been added to the results (lines 242-243).

Rev 3, comment 3: *Could you please clarify if you counted CD8⁺ T cells or Trm specifically in CXCR6 KO mice? It would be valuable to present the data on the number of Trm in CXCR6 KO T cell transfer mice over time.*

Response: As stated above, we find that the CD8⁺ T cells in the skin express T_{RM} markers (CD103+KLRG1neg, Supp. Fig 1c). We have also added flow cytometric data (Supp Fig 1a) showing that 95% of bulk CD8 T cells isolated from MAV skin express CD103 and CD69.

Rev 3, comment 4: *Was the human healthy control skin age and sex matched with the melanoma patient samples?*

Response: Because the healthy control skin was de-identified, we are unable to determine the sex and age of these donors. This information has been added to the methods (line 347).

Rev 3, comments 5 & 6: *Fig 2D were the Trm in human MAV only in hair follicles, or in the inter-follicular spaces? Please include a discussion point describing the differences between location in human and mouse Trm in MAV and why this might occur (i.e. pigmentation in B6 mouse skin is primarily derived from hair/hair follicles whereas humans have inter-follicular melanocytes). Fig 3D please provide orientation/labeling on the histology for the human samples. Are these CXCL16⁺ APC clusters interfollicular or in hair follicles? Are they proximal to the tumor site or from areas of distal depigmentation? - can you figure out where depigmentation? I would be interested (borders of white and black skin-scanned slides) to know if these are in distal depigmented areas, which would mirror the spread of depigmentation in the mice, or if they are limited to proximal sites.*

Response: We thank Rev 3 for bringing up this important point. Human skin has fewer hair follicles than mice, and melanocytes are found in both hair follicles and the interfollicular regions. The CD8 T_{RM} and CD11c cell clusters in human melanoma-associated vitiligo skin are not restricted to the hair follicles but were scattered throughout depigmented skin near pigmented skin (lines 107-112). Orientation and proximity to border of pigmented and depigmented skin, identified by Fontana Masson staining have been added to new Supp. Fig. 3. At the time of analysis, melanoma associated vitiligo patients were in complete remission and do not have tumors. Harvested skin is within areas of distal depigmentation and not proximal to the original tumor site.

Rev 3, comments 7 & 8: *Fig 4G & 5F how many of these CD8 T cells express TRM phenotypic markers? Fig 6B & C, the authors use these data to support their conclusion that CXCR6 is required for Trm maintenance. However, the data could be interpreted as CXCR6/CXCL16 is required for lesion spread (i.e. establishment of new lesions and formation of new Trm). This is a major difference in my interpretation of the authors' data versus how it is presented in this manuscript.*

Response: See response to concerns 2 and 3 above.

Rev 3, comment 9: *Fig S1C compares Pmel to endogenous CD8 T cells and concludes that differences represent Trm vs other T cell phenotypes. However, the authors present data in Fig 1 (and in previous publications from Turk lab) showing MAV without Pmel transfer, so presumably some of these endogenous CD8 T cells are in fact antigen-specific. It would be interesting to know if the*

endogenous antigen-specific Trm share similar transcriptional profiles or if the Pmel cells are unique, which might be due to their TCR, the fact that they are transgenic, etc.

Response: All experiments presented in this study besides those in Fig 5a-d and Supp. Fig. 1c show MAV without Pmel transfer. When we transfer a small number of Pmel T cells, they act as a tracker population that differentiate into TRM cells along with 95% of the bulk CD8 T cells in MAV skin (Supp Fig 1a). Bulk and Pmel T cell populations do share the same T_{RM} transcriptional signature, although Pmel cells express this signature to a higher degree (expression is normalized in each row). This is likely because the bulk CD8 T_{RM} cells are a mixed population. In a separate manuscript recently accepted at *Immunity*, we found that 1 - 5.5% of the bulk skin CD8 T_{RM} cells are specific for the melanocyte antigen Tyrosinase Related Protein2 (TRP2, aa 180-188). We strongly agree that it will be interesting to determine whether antigen specificity/affinity contributes to differential T_{RM} cell transcription. This is an active area of current and future investigation in our labs that requires single cell RNAseq analysis in combination with TCR sequencing and/or tetramer staining. We hope to include these data in future reports that will address whether TCR signals instruct T_{RM} cell differentiation and maintenance.

Rev 3, comment 10: *Other type of immune cells including B cells (B220), T cells (co-stain), and NK cells also express CD11c particularly in inflammatory conditions. Can the authors provide further justification for calling these DCs?*

Response: We find very few B cells and CD4 T cells in the MAV CD8 aggregates. IHC images for B cells (B220) and CD4 T cells have been added to Supp. Fig. 2a, b.

Rev 3, comment 11: *to prove tethering, MP-IVM (in vivo microscopy) could have been conducted. I understand this is a very costly, technically-challenging experiment, so for this study I would not require this type of analysis. The authors do show clustering/distance by static microscopy, but I just want to note here that this is a limitation for interpretation.*

Response: We thank Rev 3 for making this point and for realizing the technical challenges of MP-IVM, which are beyond our current capabilities. We have, however, softened the language in the text regarding the tethering/firm adhesion function of the CXCR6-CXCL16 axis by also discussing an alternative role for CXCR6-CXCL16 in recruitment of CD8 T_{RM} cells to vitiligo skin (lines 77, 64-68, 315-316).

Rev 3, other concerns:

- *Please clarify that Fig 3a is from mouse?* **Response:** Fig 3a is from mouse; this is now noted.
- *Fig S1 figure legend refers to (D) but not (C) heatmap. Please correct this.* **Response:** corrected
- *P11 after ref to (Fig 5d) is a sentence fragment, suggest removing "While".* **Response:** corrected
- *P15 discussion the phrase "yet, the underlying factors that trigger their formation and persistence remain unclear." Is redundant w previous sections, suggest to remove.* **Response:** removed
- *P16 "tempting to consider" is a colloquialism, suggest rewording.* **Response:** reworded
- *P16 "It is interesting to note that in our mouse model, MAV initially develops at the site of tumor excision and other sites with minor skin abrasions (e.g. ear tag); however, a mechanistic understanding of this effect remains unclear" This is called koebnerization; both melanoma and vitiligo can exhibit koebner phenomenon. I suggest reading/citing some of the following literature: G Weiss et al JEADV 2002, Sagi & Trau Clinics in Derm 2011, N van Geel et al Clin Lab Invest 2012, Khurram et al Ann Dermatol 2017, Larsabal et al JAAD 2017, R C Liu et al Clin Exp Dermatol 2019, Redondo et al Derm Surgery 2006, Federica et al Dermatologic Surgery*

2009, **Response:** This is an interesting point. Our reading of these citations indicates that the mechanism driving koebnerization remains unclear. However, we have now added text and citations for Koebner phenomenon.

- P22 "*All performed by Dartmouth's Pathology Shared Resource*" is a sentence fragment, suggest adding "*immunohistochemistry procedures were*". **Response:** corrected

July 14, 2021

RE: Life Science Alliance Manuscript #LSA-2021-01056-TR

Dr. Yina Hsing Huang
Geisel School of Medicine
Pathology, Microbiology and Immunology
1 Medical Center Dr
Lebanon, NH 03756

Dear Dr. Huang,

Thank you for submitting your revised manuscript entitled "Dendritic cells maintain anti-tumor immunity by positioning CD8 skin resident memory T cells". We would be happy to publish your paper in Life Science Alliance pending final revisions necessary to meet our formatting guidelines, as well as revisions in response to the Reviewers' remaining points.

- please add a Category for your manuscript in our system
- please note that the title in the system and the manuscript file must match
- please make sure the author order in your manuscript and our system match
- please be sure that you inserted all Authors in the Authors Contribution section
- please use capital letters when introducing panels in the figure legends (i.e. instead of a please use A, etc...) and also as callouts through the manuscript text
- please add callouts for Figure S2A, B to your main manuscript text
- please add scale bars to Figure S3, and indicate their size in the Legend
- please provide a Data Availability Statement if the RNA-seq has been deposited somewhere
- please include a study approval statement to the human tissue samples section of the Materials & Methods

LSA now encourages authors to provide a 30-60 second video where the study is briefly explained. We will use these videos on social media to promote the published paper and the presenting author. Corresponding or first-authors are welcome to submit the video. Please submit only one video per manuscript. The video can be emailed to contact@life-science-alliance.org

A. FINAL FILES:

B. MANUSCRIPT ORGANIZATION AND FORMATTING:

Sincerely,

Reviewer #1 (Comments to the Authors (Required)):

The authors have satisfactorily addressed all the issues raised by this reviewer. In particular, the presumptive APC are described throughout as CD11c+ cells rather than dendritic cells, and the authors include a detailed discussion based on transcriptional profiles to explain their identification of at least the CD11c+ CD11b- populations as cDC1. There are still a very few places towards the beginning of the manuscript where "dendritic cells" rather than "CD11c+ cells" is used, and this could perhaps be corrected.

This reviewer agrees that FMO samples are the best way (in the absence of KO cells) to determine autofluorescence.

The figures mentioned in the original review are much improved, and substantially increase their impact.

Concerns raised by other reviewers have also been addressed very thoroughly.

Reviewer #2 (Comments to the Authors (Required)):

The authors have satisfactorily addressed my various concerns

Reviewer #3 (Comments to the Authors (Required)):

Thank you for addressing my comments and concerns. The additional immunohistochemistry stains and supplemental figures describing T_{reg} help to make the data more clear.

There is a white box covering part of your IHC image in the CD11c CD8 panel (top right) for the MAV human skin supplemental figure that should be removed.

July 23, 2021

RE: Life Science Alliance Manuscript #LSA-2021-01056-TRR

Dr. Yina Hsing Huang
Geisel School of Medicine
Pathology, Microbiology and Immunology
1 Medical Center Dr
Lebanon, NH 03756

Dear Dr. Huang,

Thank you for submitting your Research Article entitled "Dendritic cells maintain anti-tumor immunity by positioning CD8 skin resident memory T cells". It is a pleasure to let you know that your manuscript is now accepted for publication in Life Science Alliance. Congratulations on this interesting work.

DISTRIBUTION OF MATERIALS:

Again, congratulations on a very nice paper. I hope you found the review process to be constructive and are pleased with how the manuscript was handled editorially. We look forward to future exciting submissions from your lab.

Sincerely,
